# DMM: Building a Versatile Image Generation Model via Distillation-based Model Merging

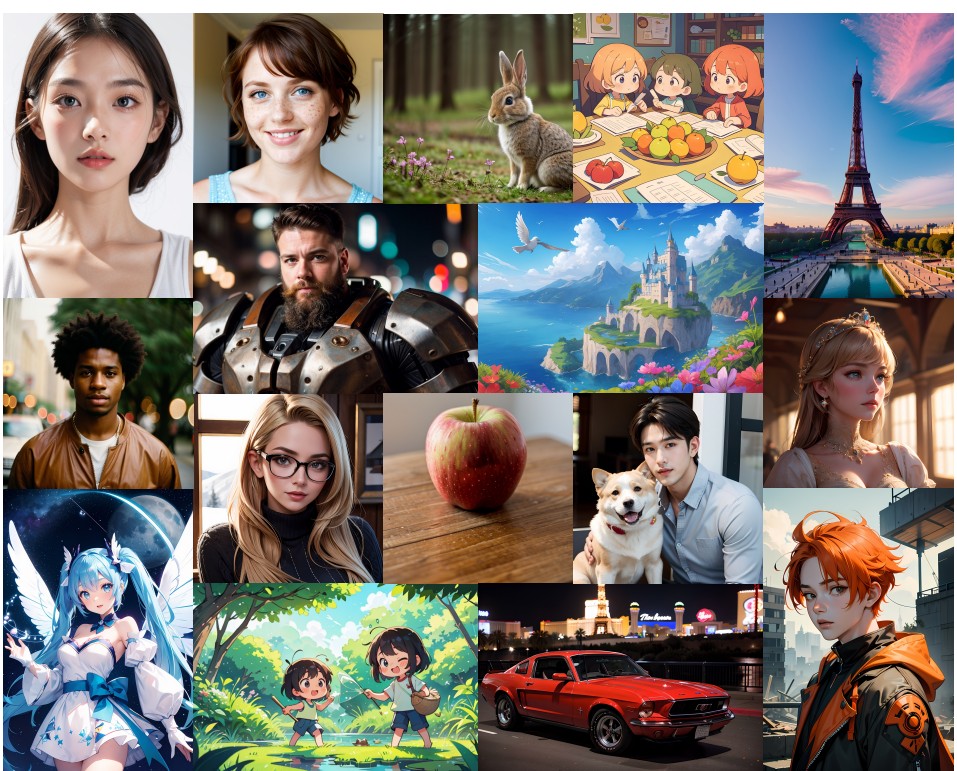

Figure 1: **Examples of image generation.** Our single model is able to generate images with various expert styles (realistic style, Asian portrait, anime style, etc.) under the control of style prompts.

## Abstract

The success of text-to-image (T2I) generation models has spurred a proliferation of numerous model checkpoints fine-tuned from the same base model on various specialized datasets. This overwhelming specialized model production introduces new challenges for high parameter redundancy and huge storage cost, thereby necessitating the development of effective methods to consolidate and unify the capabilities of diverse powerful models into a single one. A common practice in model merging adopts static linear interpolation in the parameter space to achieve the goal of style mixing. However, it neglects the features of T2I generation task that numerous distinct models cover sundry styles which may lead to incompatibility and confusion in the merged model. To address this issue, we introduce a style-promptable image generation pipeline which can accurately generate arbitrary-style images under the control of style vectors. Based on this design, we propose the score distillation based model merging paradigm (DMM), compressing multiple models into a single versatile T2I model. Moreover, we rethink and reformulate the model merging task in the context of T2I generation, by presenting new merging goals and evaluation protocols. Our experiments demonstrate that DMM can compactly reorganize the knowledge from multiple teacher models and achieve controllable arbitrary-style generation.

## 1 INTRODUCTION

Diffusion models (Ho et al., 2020; Song et al., 2020b) have steadily emerged as the predominant methods in text-to-image (T2I) generation task. Thanks to the open-source base models (Rombach et al., 2022; Podell et al., 2023; Esser et al., 2024), tool libraries (von Platen et al., 2022; AUTO-MATIC1111, 2022), and communities (Civitai; LibLib), this field has achieved great progress and numerous powerful diffusion models are released. These creation platforms make it handy for developers to fine-tune the powerful base models such as Stable Diffusion V1.5 on their specialized datasets to achieve customized generation with various styles and then upload and share the models. In spite of this fast development and prosperity, we are facing some dilemmas. First, for the thousands of personalized models, each contains billions of parameters and is saved as a large checkpoint binary file, leading to serious parameter waste and storage overhead. Second, due to limited data size and computational resources, these models are typically trained to achieve expertise in certain style domains and fail to cover a wide range of scenarios in the world. It brings many inconveniences in practical deployment when it requires different specific styles. For example, in commercial applications, each model needs to be deployed as an independent service on GPU clusters, which leads to the high overhead of computing resources due to the large model size. For personal users, when switching among different models, the disk and memory loading is also time-consuming, affecting the efficiency and experience. These issues can be alleviated if we can unify these expertise of different expert models into a single one. Currently, it is very challenging to build a versatile model, which is able to cover the knowledge of different models and supports steerable inference to accurately generate arbitrary-style images.

Model merging (Yang et al., 2024; Tang et al., 2024) is a technique that attempts to mitigate the above problems. It has shown efficacy in many fields such as large language model (LLM), but has not been deeply discussed in the T2I generation task. The existing prevalent practice for T2I diffusion models is to apply static weighted merging of model parameters (Weighted-Merging), to achieve style mixing and enhance the outputs. However, this merging method still has some critical issues. First and foremost, this approach limits the range of source models to similar domains, because directly merging models of various styles will cause conflict and style confusion. For example, for a realistic-style merged model, if we continue to merge an animation-style model into it, it will be ambiguous with different patterns and output unexpected results. Additionally, since the merge weights are usually manually set or by brute force search to obtain the best performance and parameters are statically fixed once the training is over, this scheme lacks flexibility in the style control during inference.

Therefore, we should rethink model merging in the context of T2I diffusion models and design more reasonable goals and methods. As mentioned above, one key feature we should pay attention to is there exist numerous distinct models based on diverse user creativity for different visual styles. This is relatively rare in other machine learning task territories, so direct parameter merging is insufficient to meet real application requirements. Instead, we need to devise innovative and specialized solutions to address the problem. According to our analysis, we summarize the requirements of a model merging system in the field of T2I generation: i) The merged model should preserve distinct capabilities of each source model without ambiguity, thereby truly attaining the substitution of multiple models with a singular and minimizing parameter redundancy and inefficiency. ii) The merged model should have a versatile and controllable inference mechanism to harness the knowledge of different domains, thus facilitating diverse stylistic generation and possibly generalizing to combination functionalities. iii) The training pipeline should be sustainable and scalable, supporting continual learning for new models to be incrementally merged.

Based on the above analysis, we introduce a style-promptable image generation pipeline which can generate arbitrary-style images under the control of style vectors. Based on this style-promptable generation pipeline, we resort to knowledge distillation (Hinton, 2015) and present a **d**istillation-based **m**odel **m**erging paradigm, abbreviated as **DMM**. As far as we know, we are the first to leverage knowledge distillation for the T2I diffusion model merging and building a versatile style-promptable T2I model. Additionally, we present a quantitative metric FIDt and validate our approach with the public state-of-the-art T2I models like the family of Stable Diffusion (Rombach et al., 2022; Podell et al., 2023). By merging eight different models, we train a versatile model that captures the capabilities of various styles concurrently with FIDt 77.51, while maintaining comparable performance of style mixing to previous merging methods.

To sum up, our contributions can be summarized as two folders:

- We deeply analyze the model merging task in the field of text-to-image generation and re-think a new setting of model merging task objectives with practical value. A corresponding benchmark and quantitative metric are also proposed to measure the performance of model merging under this new setting.
- We propose a new style-promptable T2I generation pipeline, which is simple to implement, steerable to different styles, and extensible to new expert models. We design the first distillation-based model merging paradigm to unify the multiple expertise into a single versatile model, supporting flexible style control mechanisms during inference.

## 2 RELATED WORK

### 2.1 DIFFUSION MODELS

Diffusion models (Ho et al., 2020; Balaji et al., 2022; Song et al., 2020b; Nichol & Dhariwal, 2021; Sohl-Dickstein et al., 2015; Karras et al., 2022) have gradually become a fundamental approach in the field of generative modeling, surpassing preceding methods in the generation of diverse and high-fidelity images. Song et al. (2020b) describes the generation process from the continuous-time perspective with stochastic differential equations (SDE), which iteratively denoise an initial noise leveraging the learned *score* of the data distribution to steer the process toward real data points. Injecting text conditions into the denoising procedure provides a more natural and user-friendly way to control image generation (Rombach et al., 2022; Balaji et al., 2022; Nichol et al., 2021; Ramesh et al., 2022). LDM (Rombach et al., 2022) performs the generation process in latent space to reduce computational costs and become the prototype of the most widespread application model Stable Diffusion (SD), which facilitates the nature of open-source AI generative models and spawned hundreds of other models and innovations worldwide. These powerful community models are typically fine-tuned on specialized datasets, thus yielding distinct experts of different style domains.

### 2.2 MODEL MERGING

Model merging is an effective technique that merges the parameters of multiple separate models with different capabilities to facilitate knowledge fusion and build a universal model. Despite its relative novelty, the field of model merging is experiencing rapid advancement and has been successfully applied across various domains (Tang et al., 2024; Yang et al., 2024). From the perspective of methodology, model merging can be implemented through linear interpolation in parameter space (Wortsman et al., 2022; Ilharco et al., 2022; Yadav et al., 2023), leveraging mode connectivity (Frankle et al., 2020), aligning features or parameters (Ainsworth et al., 2022) and ensemble distillation (Wan et al., 2024). With the emergence of large foundation models including large language models (LLM, Achiam et al. (2023); Bai et al. (2023a); Zhao et al. (2023)) and multi-modal large language models (MLLM, Yin et al. (2023); Bai et al. (2023b)), the model merging method is also explored to improve performance and efficiency (Goddard et al., 2024).

In the field of text-to-image (T2I) generation, the potential of model merging has not been fully investigated. The popular practice in the community is to apply the weighted sum of parameters of multiple models (Weighted-Merging; Chilloutmix; Majicmix), to achieve the effect of style mixing. As Biggs et al. (2024) analyzed, linearly merged diffusion models that have been fine-tuned on distinct stylized data fragments, can generate hybrid styles in a zero-shot learning context. Li et al. (2024) improves the faithfulness of T2I models by merging multiple skill-specific experts trained on synthesized datasets. Additionally, some recent works leverage ensemble learning to fuse multiple models. Wang et al. (2024a) propose an ensemble method, Adaptive Feature Aggregation (AFA), which dynamically adjusts the contributions of multiple models at the feature level according to various states, to enhance the generation quality. Nair et al. (2024) also proposes a strategy of combining aligned features of multiple models, to handle different modality conditions. However, the ensemble approaches should involve multiple models simultaneously during inference, which is computationally and memory expensive. Besides, the disposal and simple merging mechanism limit the model candidates to similar style domains, since injecting completely different models will result in mode shift and confusion. As far as we know, we are the first to leverage distillation training to merge T2I diffusion models and support flexible handling of various styles.

## 3 METHOD

### 3.1 PRELIMINARY

A generative model is usually used to represent a cerntain data probability distribution $p_{\text{data}}(\mathbf{x})$. Song (Song & Ermon, 2019; Song et al., 2020b) proposed score-based generative modeling methods, whose key idea is to model the **score** function, which is defined as the gradient of the log probability density function: $\nabla_{\mathbf{x}} \log p(\mathbf{x})$. To alleviate the difficulty of accurate score estimation in regions of low data density, we can perturb data points with noise and train score-based models on the noisy data $p_t(\mathbf{x}_t)$ instead (Song & Ermon, 2019).

Leveraging the score function, the forward perturbation and backward sampling processes can be described as stochastic differential equations (SDE) (Song et al., 2020b):

$$\text{forward-time SDE:} \quad \mathrm{d}\mathbf{x}_t = \mathbf{f}(\mathbf{x}_t, t)\mathrm{d}t + g(t)\mathrm{d}\mathbf{w}, \tag{1}$$

$$\text{reverse-time SDE:} \quad \mathrm{d}\mathbf{x}_t = \left[\mathbf{f}(\mathbf{x}_t, t) - g(t)^2 \nabla_{\mathbf{x}_t} \log p_t(\mathbf{x}_t)\right] \mathrm{d}t + g(t)\mathrm{d}\bar{\mathbf{w}}, \tag{2}$$

where $\mathbf{f}(\cdot, \cdot)$ and $g(\cdot)$ denote the drift and diffusion coefficients respectively, and $\mathbf{w}, \bar{\mathbf{w}}$ are the standard Wiener process. Moreover, a good property of this system is that there exists an ordinary differential equation (ODE), whose trajectories share the same marginal probability densities $p_t(\mathbf{x}_t)$ as the SDE, dubbed the *Probability Flow* (PF) ODE:

$$\mathrm{d}\mathbf{x}_t = \left[\mathbf{f}(\mathbf{x}_t, t) - \frac{1}{2}g(t)^2 \nabla_{\mathbf{x}_t} \log p_t(\mathbf{x}_t)\right] \mathrm{d}t. \tag{3}$$

Once we have trained a time-dependent score-based model $\mathbf{s}(\mathbf{x}_t, t; \boldsymbol{\theta}) \approx \nabla_{\mathbf{x}_t} \log p_t(\mathbf{x}_t)$, this is an instance of a neural ODE and clean images can be generated through solving it.

For training the score-based models, we can minimize the Fisher divergence between the model and the data distributions, which yields the score matching objective:

$$\boldsymbol{\theta}^* = \arg\min_{\boldsymbol{\theta}} \mathbb{E}_t \mathbb{E}_{\mathbf{x}_t \sim p_t(\mathbf{x}_t)} ||\mathbf{s}(\mathbf{x}_t, t; \boldsymbol{\theta}) - \nabla_{\mathbf{x}_t} \log p_t(\mathbf{x}_t)||. \tag{4}$$

Since the regression target $\nabla_{\mathbf{x}_t} \log p_t(\mathbf{x}_t)$ is not tractable directly, many techniques (Hyvärinen & Dayan, 2005; Vincent, 2011; Song et al., 2020a) have been explored for optimizing score matching objectives. For example, denoising score matching (Vincent, 2011) provides an equivalent but tractable optimization objectives:

$$\boldsymbol{\theta}^* = \arg\min_{\boldsymbol{\theta}} \mathbb{E}_t \mathbb{E}_{\mathbf{x}_0 \sim p_0(\mathbf{x}_0)} \mathbb{E}_{\mathbf{x}_t \sim p_t(\mathbf{x}_t|\mathbf{x}_0)} ||\mathbf{s}(\mathbf{x}_t, t; \boldsymbol{\theta}) - \nabla_{\mathbf{x}_t} \log p_t(\mathbf{x}_t|\mathbf{x}_0)||. \tag{5}$$

This objective is consistent with Denoising Diffusion Probabilistic Model (DDPM, Ho et al. (2020)). Specifically, under the formulation of DDPM with noise schedule give by $\bar{\alpha}_t$:

$$\mathbf{x}_t = \sqrt{\bar{\alpha}_t}\mathbf{x}_0 + \sqrt{1 - \bar{\alpha}_t}\boldsymbol{\epsilon}, \quad \boldsymbol{\epsilon} \sim \mathcal{N}(\mathbf{0}, \mathbf{I}), \tag{6}$$

where a denoising model $\boldsymbol{\epsilon}(\mathbf{x}_t, t; \boldsymbol{\theta})$ is trained to predict the added noise of a noisy image $\mathbf{x}_t$ with the following objective:

$$\boldsymbol{\theta}^* = \arg\min_{\boldsymbol{\theta}} \mathbb{E}_t \mathbb{E}_{\mathbf{x}_0 \sim p_0(\mathbf{x}_0)} \mathbb{E}_{\boldsymbol{\epsilon} \sim \mathcal{N}(\mathbf{0}, \mathbf{I})} \left[||\boldsymbol{\epsilon}(\mathbf{x}_t, t; \boldsymbol{\theta}) - \boldsymbol{\epsilon}||\right]. \tag{7}$$

### 3.2 TASK FORMULATION

Given a set of $N$ pre-trained isomorphic models $\{\mathbf{s}(\cdot; \boldsymbol{\theta}_i)\}_{i=1}^N$, each parameterized with $\boldsymbol{\theta}_i$, and each model is trained on different datasets (such as realistic style, anime style, etc.), thus modeling different data distributions. We use $\{p_0^{(i)}(\mathbf{x}_0)\}_{i=1}^N$ to represent the distributions corresponding to each model. Accordingly, each model predicts the corresponding score function: $\mathbf{s}(\mathbf{x}_t, t; \boldsymbol{\theta}_i) \approx \nabla_{\mathbf{x}} \log p_t^{(i)}(\mathbf{x}_t)$. Our target is to merge them into one single model with parameter $\boldsymbol{\theta}^*$ while preserving the knowledge and capabilities of each individual model. Specifically, with the merged model $\boldsymbol{\theta}^*$, given an style index $i$, $\mathbf{s}(\cdot, i; \boldsymbol{\theta}^*)$ should represent the corresponding data distribution $p_0^{(i)}(\mathbf{x}_0)$.

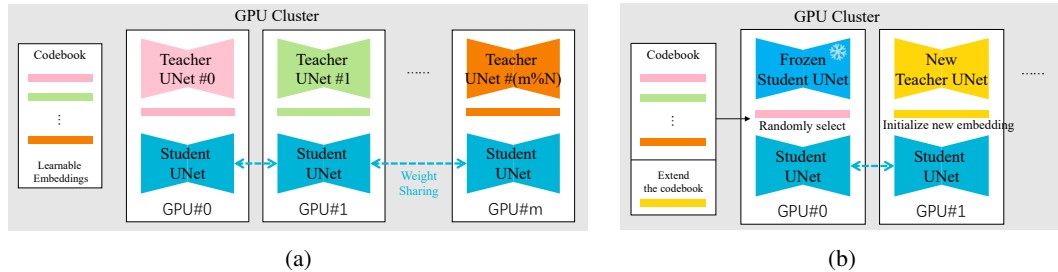

(a)                (b)

Figure 2: **Distributed Training Framework** for DMM. (a) The main model layout on a GPU cluster during training. Each node is assigned a specific teacher model to jointly supervise a student model with shared parameters. A set of learnable embeddings (style prompts) are maintained to provide hints and differentiate from each other. (b) **Continual Learning**. New teacher models are involved through initializing and adding new embeddings. The frozen pretrained student model serves as regularization with style prompts randomly selected.

One solution is to train a versatile score-based model by distilling knowledge from multiple pre-trained experts. Consequently, this gives a natural score-distillation objective:

$$\boldsymbol{\theta}^* = \arg\min_{\boldsymbol{\theta}} \sum_{i=1}^{N} \mathbb{E}_t \mathbb{E}_{\mathbf{x}_t \sim p_t^{(i)}(\mathbf{x}_t)} \left[ ||\mathbf{s}(\mathbf{x}_t, t, i; \boldsymbol{\theta}) - \nabla_{\mathbf{x}_t} \log p_t^{(i)}(\mathbf{x}_t)|| \right],$$

$$\approx \arg\min_{\boldsymbol{\theta}} \sum_{i=1}^{N} \mathbb{E}_t \mathbb{E}_{\mathbf{x}_t \sim p_t^{(i)}(\mathbf{x}_t)} \left[ ||\mathbf{s}(\mathbf{x}_t, t, i; \boldsymbol{\theta}) - \mathbf{s}(\mathbf{x}_t, t; \boldsymbol{\theta}_i)|| \right]. \quad (8)$$

Our score distillation objective resorts to the original explicit score matching objective in Eq. (4), since now we have direct access to the target score term.

### 3.3 DISTILLATION-BASED MODEL MERGING FRAMEWORK

**Distillation-based model merging.** We present a simple yet efficient distributed training framework of DMM to implement the score-distillation objective for model merging, as depicted in Fig. 2a. Our task is essentially a knowledge distillation task from multiple teacher models (Jiang et al., 2024; Gu et al., 2021; Meng et al., 2021) to a single steerable student one. Besides, we propose three types of loss functions and a data sampling strategy to boost the performance. Since it is almost impossible to load all teacher models into a single GPU's memory, we design to assign a specific teacher model to each GPU uniformly to efficiently utilize GPU memory.

**Style-promptable generation.** The student model shares the same UNet architecture as the base models, except for a few additional trainable parameters for handling the style index hint $i$ suggesting which model's style to trigger. Specifically, as shown in Fig. 2a and Fig. 3, we represent different model priors as a codebook of $N$ learnable embeddings, named *style prompts*. The implementation details of injecting style prompts are stated in Appendix A.1. These prompts are used to specify the image style and modulate the student UNet to mimic the corresponding teacher model. Once the training is finished, the style prompts provide a flexible way to control the styles at test time, which will be discussed in Sec. 4.5.

### 3.4 LOSS FUNCTION AND DATA SAMPLING

**Score distillation.** As analyzed in Sec. 3.2, we apply score distillation loss to learn different target probability distributions. Drawing the connection between DDPM and Score Matching, we can directly perform mean square error (MSE) loss on the outputs, of the $\boldsymbol{\epsilon}$-Prediction parameterized models (such as Stable Diffusion (Rombach et al., 2022)).

$$\mathcal{L}_{\text{score}}(\mathbf{x}_t, t) = \sum_{i=1}^{N} ||\boldsymbol{\epsilon}(\mathbf{x}_t, t, i; \boldsymbol{\theta}) - \boldsymbol{\epsilon}(\mathbf{x}_t, t; \boldsymbol{\theta}_i)||_2^2. \quad (9)$$

**Feature imitation.** According to the observation that many previous works (Wang et al., 2024b; Ye et al., 2023) have explored, the intermediate features of the model contain rich style informa-

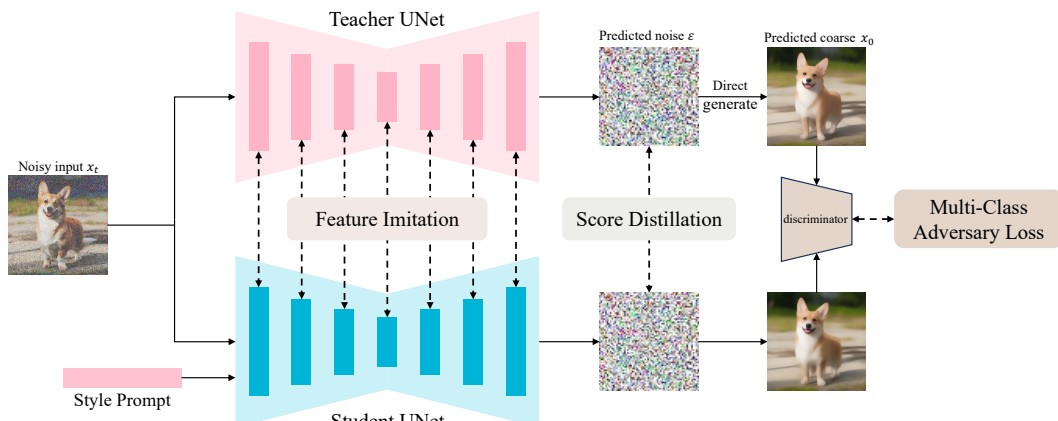

Figure 3: **Style-promptable generation pipeline for disitllation-based model merging**. Our proposed distillation objective incorporates three loss terms: Score Distillation, Feature Imitation, and Multi-Class Adversarial Loss.

tion. Therefore, we leverage feature supervision to facilitate knowledge transfer and style learning. Formally, let $\mathbf{F}_j^{\mathrm{S}}, \mathbf{F}_j^{\mathrm{T}} \in \mathbb{R}^{h \times w \times c}$ denote the feature map from student and teacher models, and the subscript represents the index of model layers. For feature imitation, we apply MSE loss:

$$\mathcal{L}_{\text{feat}}(\mathbf{x}_t, t) = \sum_{i=1}^{N} \sum_{j \in \mathcal{M}} \left\| \mathbf{F}_j^{\mathrm{S}}(\mathbf{x}_t, t, i; \boldsymbol{\theta}) - \mathbf{F}_j^{\mathrm{T}}(\mathbf{x}_t, t; \boldsymbol{\theta}_i) \right\|_2^2, \tag{10}$$

where $\mathcal{M}$ is the set of layer indices that need to be supervised. Our experiment results demonstrate that naively supervising all the layers' features can significantly boost performance.

**Multi-class adversarial loss.** To further enhance the model's ability to discriminate and fit different data distributions, we incorporate an additional GAN objective into our training framework. Generative Adversarial Networks (GAN, Song & Ermon (2019)) implicitly model the real data distribution by training a generator and a discriminator competitively. Essentially, it is optimizing Jensen–Shannon divergence (Endres & Schindelin, 2003) for matching two probability distributions. Considering we aim to learn multiple target data distributions simultaneously, we can naturally tailor a multi-class GAN to substitute the vanilla binary GAN. Specifically, given $N$ teacher models, the total number of classification heads is $2N$, where the first $N$ classes represent $N$ target styles and the last $N$ classes represent fake ones. The discriminator is trained to not only distinguish between real and fake images but also to distinguish different style distributions. This yields our multi-class adversarial loss as below:

$$\mathcal{L}_{\text{adv}}(\mathbf{x}_t, t) = -\sum_{i=1}^{N} \left[ \log \mathcal{D}_i(\mathbf{g}(\mathbf{x}_t, t, i; \boldsymbol{\theta})) + \log \mathcal{D}_{N+i}(\mathbf{g}(\mathbf{x}_t, t; \boldsymbol{\theta}_i)) \right], \tag{11}$$

where $\mathcal{D}_i(\cdot)$ is the discriminator predicted probabilities for the $i$-th class, and $\mathbf{g}$ is the generation function for sampling clean images from model outputs. To sample efficiently, we leverage the formulation of predicting $\mathbf{x}_0$ directly from $\mathbf{x}_t$ and $\boldsymbol{\epsilon}(\mathbf{x}_t, t; \boldsymbol{\theta})$ according to the noise scheduler as Eq. (6) (Xu et al., 2024b; Lu et al., 2023):

$$\mathbf{g}(\mathbf{x}_t, t; \boldsymbol{\theta}) = \frac{1}{\sqrt{\bar{\alpha}_t}} \left( \mathbf{x}_t - \sqrt{1 - \bar{\alpha}_t} \boldsymbol{\epsilon}(\mathbf{x}_t, t; \boldsymbol{\theta}) \right). \tag{12}$$

**Training data sampling.** Herein, there is still a challenge in that we do not have direct access to the original training data of each teacher model, thus unable to sample training data points $\mathbf{x} \sim p_t^{(i)}(\mathbf{x}_t)$. Fortunately, due to the good property of direct access to the score target under distillation, this optimization process can be considered a conventional regression task, and the training data can be generalized to a common dataset. As depicted in Fig. 3, the final optimization objective is the weighted combination of the three loss terms as below:

$$\mathcal{L}_{\text{total}} = \mathbb{E}_t \mathbb{E}_{\mathbf{x}_t \sim p_t(\mathbf{x}_t)} \left[ \mathcal{L}_{\text{score}} + \lambda_{\text{feat}} \mathcal{L}_{\text{feat}} + \lambda_{\text{adv}} \mathcal{L}_{\text{adv}} \right]. \tag{13}$$

Our experimental results show that sampling from a general common training dataset is enough to effectively distill knowledge from different teacher models. The intuition behind this is since we train the model on noise-perturbed data, the different noisy data distributions overlap with each other, especially at large timesteps. To compensate for regions of low data density at small timesteps, we use teacher models to synthesize a small number (hundreds) of images and fine-tune the model with a few thousand iterations. This stage can further refine the generation quality and is highly efficient, consuming only a few GPU hours.

### 3.5 Incremental Learning with Regularization

Our proposed distillation framework supports flexible incremental learning to merge new models into the current checkpoint. In practice, we only need to extend the set of style prompts by adding new ones (randomly initialized), then fine-tune the model with supervision as discussed in Sec. 3.4 for them. However, only optimizing for new targets can lead to the problem of catastrophically forgetting existing knowledge. To alleviate this issue, we propose an efficient incremental learning approach with regularization as illustrated in Fig. 2b. We adopt a self-supervised approach to preserve the knowledge of the learned models to achieve regularization. Specifically, we treat the pretrained merged model as the teacher with its parameters frozen, to supervise the student model with corresponding style prompt inputs. The style prompt indices are randomly selected in each training iteration. This method allows for efficient device resource consumption that assigning one GPU is enough for regularization, instead of deploying $N$ GPUs for all old teachers.

## 4 Experiments

### 4.1 Evaluation

Before presenting the experiments, we first introduce the evaluation protocol under our task setting. To quantitatively measure the performance of the model for learning different target model distributions, we propose an evaluation metric based on Fréchet inception distance (FID). The FID (Heusel et al., 2017) metric measures the distance between two probability distributions, the distributions of model predictions and reference images. Consequently, we sample images through the student model with different $N$ style prompts and $N$ different teacher models and calculate the FID score pairwisely, which gives an FID matrix $\mathbf{M} \in \mathbb{R}^{N \times N}$ satisfying:

$$\mathbf{M}(i,j) = \text{FID}\left(\hat{P}_{\text{S}}^{(i)}, P_{\text{T}}^{(j)}\right), \tag{14}$$

where $P_{\text{S/T}}^{(i)}$ represents the distribution of generated images of student/teacher models with the $i$-th style. Ideally, we hope the FID scores on the diagonal to be as small as possible, which indicates how well the model matches different target distributions. Therefore, we define the metrics **FIDt** as the trace of $\mathbf{M}$ and use it to observe the performance of model merging:

$$\text{FIDt} := \text{Tr}(\mathbf{M}) = \sum_{i=1}^{N} \mathbf{M}(i,i). \tag{15}$$

Besides, we leverage the teacher models to sample two batches of images with different random seeds and calculate their FID matrix $\mathbf{M}_{\text{ref}}$ as above. We consider the $\mathbf{M}_{\text{ref}}$ as a reference since it suggests the upper bound of model performance. In this paper, we sample 5k images per batch using text prompts from MS-COCO (Lin et al., 2014) validation set for FID calculation.

### 4.2 Implementation Details

Our main experiments are based on SDv1.5 architecture and the student model is initialized from SDv1.5 weights. For base models to be merged, we select eight popular models with different styles from open-source model communities. We leverage JourneyDB (Sun et al., 2024) as our training dataset. The distillation training is conducted on 16 A100 GPUs, and the batch size is 320 with each GPU holding 20 samples. We train the model for 100k iterations, which takes about 32 GPU days. More implementation details are provided in the Appendices. We also conduct SDXL architecture experiments, and report results in the Appendices.

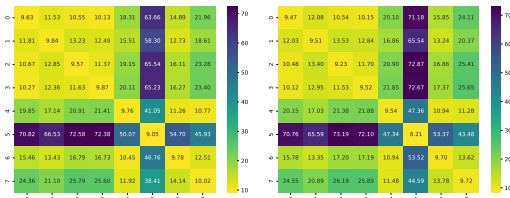

Figure 4: Heatmap of the FID matrix. The left one is the result $\mathbf{M}$ of our model, and the right one is the reference matrix $\mathbf{M}_{\text{ref}}$.

| Methods | FIDt↓ |
|---|---|
| + Score Distillation | 80.69 |
| + Feature Imitation | 79.27 |
| + Multi-Class Adversarial | 78.38 |
| + Synthesized Finetune | **77.51** |
| Teacher Reference | 74.91 |

Table 1: **Ablation Results**. The first three lines are our proposed three types of losses. The fourth line is the fine-tuning stage with synthesized data.

| # | Stage | #Model | Regular. | 1 | 2 | 3 | 4 | 5 | 6 | 7 | 8 |
|---|---|---|---|---|---|---|---|---|---|---|---|
| 0 | Teacher | 8 | - | 9.5 | 9.5 | 9.2 | 9.5 | 9.5 | 8.2 | 9.7 | 9.7 |
| 1 | Train | 4 | - | 9.5 | 9.8 | 9.5 | 9.7 | - | - | - | - |
| 2 | Train | 8 | - | 9.6 | 9.8 | 9.6 | 9.9 | 9.8 | 9.0 | 9.8 | 10.0 |
| 3 | Fine-tune | 4 + 4 | ✗ | 21.0 | 23.9 | 26.7 | 18.5 | 9.7 | 9.6 | 9.7 | 10.0 |
| 4 | Fine-tune | 4 + 4 | ✓ | 9.7 | 9.8 | 9.6 | 9.8 | 9.9 | 9.0 | 9.8 | 10.0 |

Table 2: **Incremental Learning**. Each column is Experiment Number, Training Stage, Number of Merged Models, Regularization, and FID scores on the diagonal of $\mathbf{M}$. 'Teacher' represents the reference upper-bound results of teacher models. 'Train' denotes the model is trained from the start, 'Fine-tune' denotes the model is fine-tuned from the checkpoint of #1.

### 4.3 MAIN RESULTS

The FID score matrix $\mathbf{M}$ and reference matrix $\mathbf{M}_{\text{ref}}$ are illustrated in Fig. 4, and the corresponding FIDt metric is presented in Tab. 1, where the default final result is highlighted in gray . Compared to the reference FIDt of 74.91, our approach achieves 77.51 FIDt which is quite close to the upper bound. Besides, we can observe that our result matrix $\mathbf{M}$ shows similar patterns to $\mathbf{M}_{\text{ref}}$. This demonstrates our method's effectiveness in matching all different target model distributions, achieving *all-in-one* functionality, and contributing to a versatile model.

For visual qualitative results, we show generated images from our model with different style prompts in Fig. 6. We can see that our model can synthesize images that are highly consistent with the teacher models. This further illustrates that the domain knowledge and capabilities of different models have been compactly merged into one single model, thus significantly reducing parameter redundancy.

### 4.4 ABLATION STUDY

**Loss functions and synthesized data.** Tab. 1 ablates the design of our loss functions: score distillation, feature imitation, and multi-class adversarial loss. It can be seen that feature imitation can increase the baseline performance by 1.5 FIDt, and adding multi-class adversarial loss further improves by 1.42 FIDt, which indicates the effectiveness of both feature imitation and adversarial learning for distribution matching in the model merging task. Furthermore, the fine-tuning stage utilizing synthesized data can enhance the FIDt by 0.87 at a minimal cost.

**Incremental learning with regularization.** To ascertain the efficacy of our proposed incremental learning mechanism, we conduct experiments in Tab. 2. #1 and #2 are two experiments with our method to merge four and eight models respectively, and #3,4 are experiments to fine-tune from #1 checkpoint with and without regularization to add the remaining four models. Experiment #3 shows that the first four FID scores have diverged without any regularization, revealing that the model severely suffers from catastrophic forgetting during continual learning. Experiment #4 crucially demonstrates that our regularization strategy can alleviate this phenomenon and reach parallel performance with full training, thus guaranteeing stable incremental learning.

### 4.5 EXTENSION APPLICATIONS

Apart from triggering different generation styles, our framework's significant advantage is its flexibility and scalability, which support many extension applications with excellent style control ability.

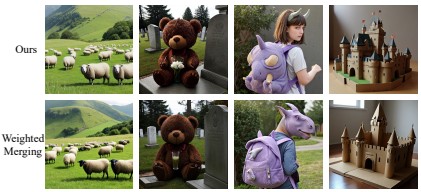

| Method | CLIP-Score | Aes-Score | Pick-Score |
|--------|-----------|-----------|-----------|
| WM | 32.47 | 5.58 | 22.36 |
| DMM | 32.51 | 5.58 | 22.35 |

Figure 5: Visualization comparison with Weighted Merging.

Table 3: Quantitative comparison with Weighted Merging on metrics of CLIP-Score (Radford et al., 2021), Aesthetic-Score (LAION-AESTHETICS), and Pick-Score (Kirstain et al., 2023). 'WM' denotes the Weighted Merging method.

**Style mixing.** Benefiting from our proposed embedding-based style prompts, our DMM is easy to tailor for style combinations during inference. Compared to the common practice of weight merging (Weighted-Merging), which manually pre-determines the weights to merge parameters of multiple SD models for mixed effects, our DMM can achieve the purpose more efficiently. Specifically, given the $N$ prior embeddings $\{\mathbf{e}_i\}_{i=1}^N$, we can represent the mixture of different model distributions through interpolation of them: $\tilde{\mathbf{e}} = \sum_{i=1}^N w_i \mathbf{e}_i$, s.t. $\sum_{i=1}^N w_i = 1$, and feed it into the model. To verify the effectiveness of our proposed embedding interpolation approach for style mixing, we conduct experiments on the first four realistic-style merged models. Specifically, for DMM we directly average the first four style prompts during inference, and for the baseline method, we merge the parameters of the four models. We compare the results on the test set of COCO30K (Lin et al., 2014), as shown in Fig 5 and Tab. 3. We can see that DMM can achieve semantically and aesthetically comparable performance of style mixing to Weighted Merging (WM), while our approach is more flexible and can afford many style generation simultaneously. Additionally, to further illustrates that our mixing strategy really take effects, we perform interpolation on two styles and adjust the weights, which delivers a smooth and stable transition between them as shown in Fig. 7. More results are provided in the Appendices.

**Compatibility with plugins.** Due to the model being trained based on Stable Diffusion, and the feature imitation module enabling the intermediate representation of hidden layers to be aligned with the base model, our DMM is seamlessly compatible with various downstream plugins such as ControlNet (Zhang et al., 2023), LoRA (Hu et al., 2021), and IP-Adapter Ye et al., 2023, without extra training. Besides, our approach can be easily adapted to techniques of integrating multiple diffusion processes and spatial control, such as Mixture-of-Diffusers (Jiménez, 2023) and MultiDiffusion (Meng et al., 2021). Since DMM can leverage different styles flexibly through our proposed style prompts, it can further boost the diversity and versatility of integrated generation. We display some results with ControlNet and IP-Adapter in Fig. 8, from which we can see that these plug-and-play modules work on different styles with only a single versatile model. For integrated generation pipelines, we show the results of DMM combined with Mixture-of-Diffusers in Fig 9, the panoramas that harmonize different styles. These extensions greatly boost the efficiency of creativity and application. More results are in the Appendices. These results show the general application potential of our DMM method.

**Transferring to distillation-based acceleration method.** Because our approach is based on distillation and the style prompt is lightweight to embed into the model, our merging mechanism can be naturally combined with many distillation-based acceleration methods (Luo et al., 2023; Xu et al., 2024a). We will illustrate the implementation and results in the Appendices.

## 5 CONCLUSION

In this paper, we have rethinked the model merging task in the realm of T2I diffusion models and built a versatile style-promptable diffusion models for steerable image generation. Specifically, we present DMM, a simple yet effective merging paradigm based on score distillation. DMM leverages three types of loss functions to boost the merging performance and perform regularization to support stable continual learning. With our designed embedding-based style control mechanism, users can operate the style prompts to execute various style combinations flexibly during inference. We design an evaluation benchmark with the new metric and the results demonstrate our merged model is able to well mimic the expert teacher model in image generation quality. We hope our DMM can facilitate the development of model merging in image generative models.

portrait photo of a girl, long golden hair, flowers, best quality

handsome boy, blue suit, upper body portrait

A white plate containing salmon, rice, and broccoli.

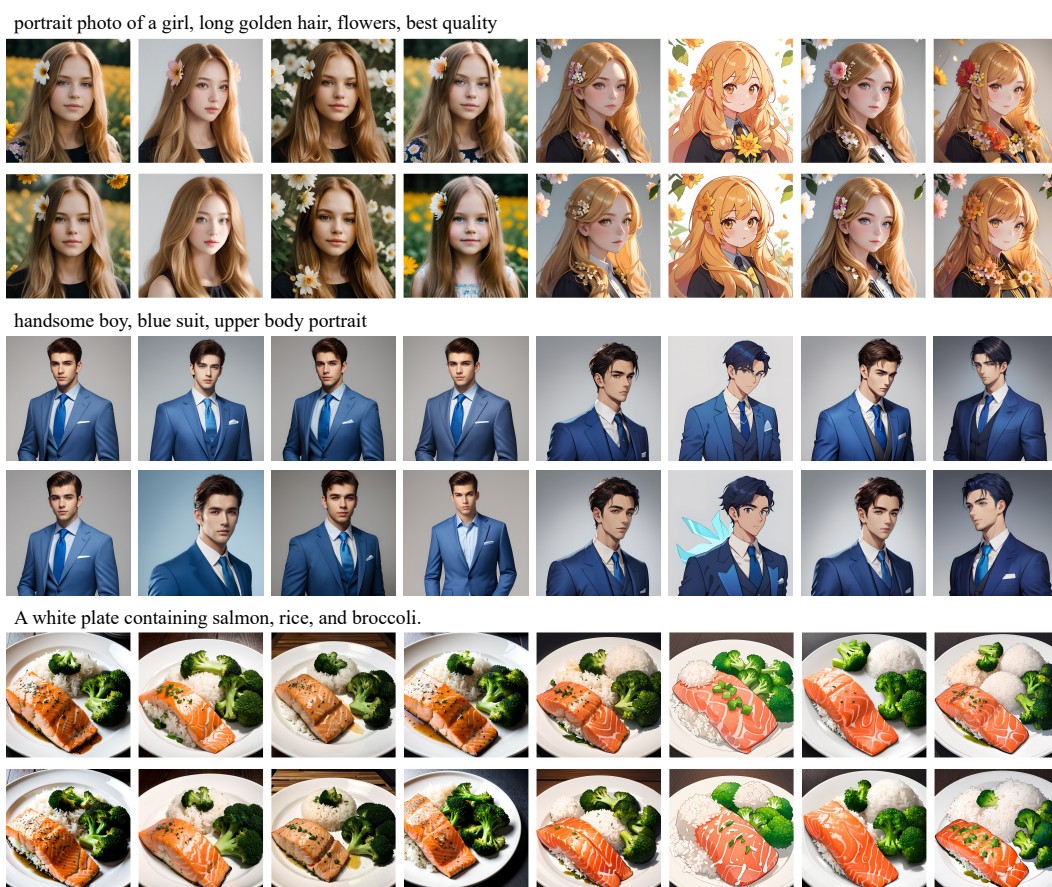

Figure 6: Visual results with different style selections. In each group, the first line is our model's results, the second is the teacher models' results. More examples are provided in the Appendices.

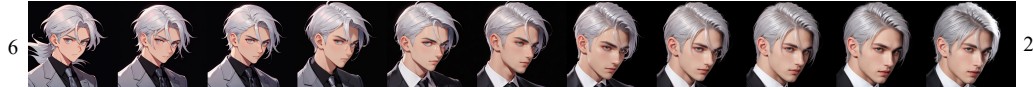

Figure 7: The results of interpolation between two styles. The number on the side is the model index. The weight list of one ingredient is [0.0, 0.1, 0.2, 0.3, 0.4, 0.5, 0.6, 0.7, 0.8, 0.9, 1.0].

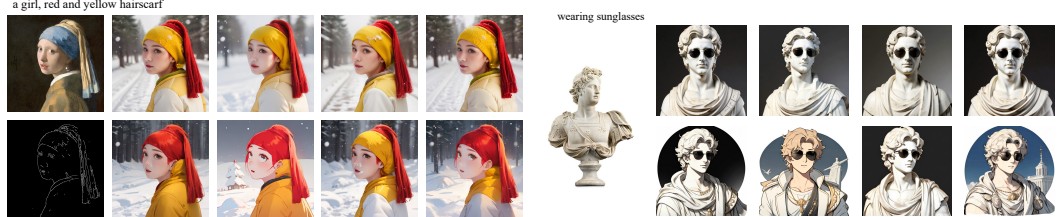

Figure 8: Visual results of DMM integrated with ControlNet-Canny and IP-Adapter.

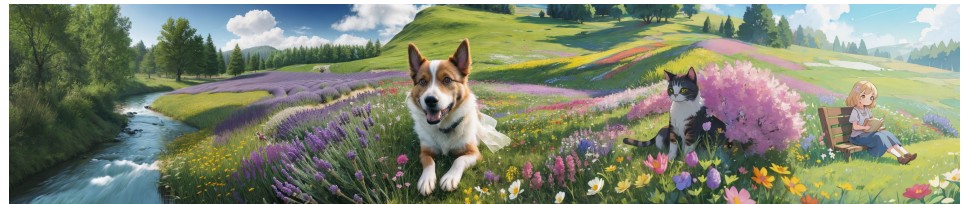

Figure 9: Visual results of DMM combined with Mixture-of-Diffusers.

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

# A    IMPLEMENTATION DETAILS

This section provides a brief overview of the implementation details.

## A.1    STYLE PROMPTS DESIGN

As proposed in Sec. 3.3, we represent the different style prompts as a set of trainable embeddings, forming a codebook. The dimension of embeddings is the same as that of timestep embedding in SDv1.5, which is 1280. These embeddings are randomly initialized before training and can be indexed to imply the mode. We adopt a simple strategy to inject the style prompts into the UNet model. Specifically, we first align the embeddings with an MLP and then add it to the timestep embedding, which is lightweight enough for plug-and-play purpose. The codebook and the MLP which contain two $\mathbb{R}^{d \times d}$ linear projections are all additional parameters.

## A.2    MULTI-CLASS GAN CLASSIFIER DESIGN

The architecture design of our multi-class GAN classifier is inspired by DMD2 (Yin et al., 2024) and SDXL-Lightning (Lin et al., 2024). Specifically, we attach a sequence of convolutions, group normalization, and SiLU activations on top of the middle block of the backbone UNet. The only difference is that the final classification projection dimension is $2N$ instead of a single scalar. Moreover, we diffuse the discriminator input images with random noise to improve the robustness.

## A.3    TRAINING SETTING

**SDv1.5**    For the main experiment on SDv1.5 architecture, we merge a group of eight popular models from the open-source platform and list them below:

| No. | Model Name | Style | Source |
|-----|------------|-------|--------|
| 1 | JuggernautReborn | Realistic | https://civitai.com/models/46422 |
| 2 | MajicmixRealisticV7 | Realistic, Asian | https://civitai.com/models/43331 |
| 3 | EpicRealismV5 | Realistic | https://civitai.com/models/25694 |
| 4 | RealisticVisionV5 | Realistic | https://civitai.com/models/4201 |
| 5 | MajicmixFantasyV3 | Anime | https://civitai.com/models/41865 |
| 6 | MinimalismV2 | Illustration | https://www.liblib.art/modelinfo |
| 7 | RealCartoon3dV17 | 3D Cartoon | https://civitai.com/models/94809 |
| 8 | AWPaintingV1.4 | Anime | https://civitai.com/models/84476 |

Table 4: The information of all models to be merged on SDv1.5.

We leverage JourneyDB (Sun et al., 2024) as our training dataset. The distillation training is conducted on 16 NVIDIA A100 GPUs, and the batch size is 320 with each GPU holding 20 samples. We use AdamW optimizer and the learning rate is $10^{-5}$, for both the diffusion model and the discriminator. We train the model for 100k iterations, which costs about 32 GPU days. The loss weights in Eq. 13 are set as $\lambda_{\text{feat}} = 0.001, \lambda_{\text{adv}} = 0.01$. To support widely used Classifier-Free Guidance (CFG, Ho & Salimans (2022)), we replace $10\%$ text embeddings with null embeddings for training the unconditional model. For the synthesized fine-tuning stage, we synthesize 1.5k images per teacher and fine-tune the model with batch size 10 and 10k iterations, costing about 16 GPU hours.

**SDXL**    We additionally conduct experiments on SDXL architecture and display the results in Sec. B. The teacher models are:

The training hyper-parameter settings are the same as SDv1.5 experiments, except that the batch size is 10 per GPU to accommodate GPU memory usage.

**SDv1.5-SPLAM**    As claimed in Sec. 4.5, our approach can be transferred to distillation-based acceleration methods and obtain a fast version of the merged model. To train an DMM-SPLAM, we initialize the model with the checkpoint of vanilla DMM and replace the loss with SPLAM loss.

| No. | Model Name | Style | Source |
|-----|-----------|-------|--------|
| 1 | JuggernautXLV10 | Realistic | https://civitai.com/models/133005 |
| 2 | LEOSAMXLV7 | Realistic | https://www.liblib.art/modelinfo |
| 3 | GhostXLV1 | Anime | https://civitai.com/models/312431 |
| 4 | AnimagineV3.1 | Anime | https://civitai.com/models/260267 |

Table 5: The information of all models to be merged on SDXL.

The batch size is 50 per GPU, and since rapid convergence, we have trained DMM-SPLAM with 8 GPUs for 6k iterations.

### A.4 DMM TRAINING ALGORITHM

We present the overall training process in Algorithm 1. We omit some insignificant parts such as operations of VAE and text condition for simplicity.

---

**Algorithm 1** DMM Training Algorithm

---

**Require:** Number of teacher models $N$. Number of GPU nodes $M$. Current GPU node index $j$. Student model $\boldsymbol{\epsilon}(\cdot; \boldsymbol{\theta})$. Teacher models $\{\boldsymbol{\epsilon}(\cdot; \boldsymbol{\theta}_i)\}_{i=1}^N$. Discriminator $\mathcal{D}(\cdot; \boldsymbol{\eta})$.

  Get the teacher index of the current node $i \leftarrow j \% N + 1$
  **repeat**
    Sample $\mathbf{x}_0$ from data distribution, $\boldsymbol{\epsilon} \sim \mathcal{N}(0, I), t \in [0, T]$
    $\mathbf{x}_t \leftarrow \texttt{add\_noise}(\mathbf{x}_0, \boldsymbol{\epsilon}, t)$
    $\boldsymbol{\epsilon}_{\text{stu}}, \mathbf{F}_{\text{stu}} \leftarrow \boldsymbol{\epsilon}(\mathbf{x}_t, t, i; \boldsymbol{\theta})$ ▷ Get model outputs and intermediate features.
    $\boldsymbol{\epsilon}_{\text{tea}}, \mathbf{F}_{\text{tea}} \leftarrow \boldsymbol{\epsilon}(\mathbf{x}_t, t; \boldsymbol{\theta}_i)$
    $\hat{\mathbf{x}}_{0,\text{stu}} \leftarrow \mathbf{g}(\mathbf{x}_t, t, \boldsymbol{\epsilon}_{\text{stu}})$ ▷ Predict the clean image directly.
    $\hat{\mathbf{x}}_{0,\text{tea}} \leftarrow \mathbf{g}(\mathbf{x}_t, t, \boldsymbol{\epsilon}_{\text{tea}})$
    $l_{\text{stu}} \leftarrow \mathcal{D}(\hat{\mathbf{x}}_{0,\text{stu}})$ ▷ Predict the logits $l \in \mathbb{R}^{2N}$.
    $\mathcal{L}_{\text{adv}} \leftarrow \texttt{cross\_entrophy}(l_{\text{stu}}, i)$
    $\mathcal{L}_{\text{score}} \leftarrow \|\boldsymbol{\epsilon}_{\text{stu}} - \boldsymbol{\epsilon}_{\text{stu}}\|_2^2$
    $\mathcal{L}_{\text{feat}} \leftarrow \|\mathbf{F}_{\text{stu}} - \mathbf{F}_{\text{stu}}\|_2^2$
    $\mathcal{L}_{\text{gen}} \leftarrow \mathcal{L}_{\text{score}} + \lambda_{\text{feat}} \mathcal{L}_{\text{feat}} + \lambda_{\text{adv}} \mathcal{L}_{\text{adv}}$
    Update $\boldsymbol{\theta} \leftarrow \boldsymbol{\theta} - \frac{\partial}{\partial \boldsymbol{\theta}} \mathcal{L}_{\text{gen}}$ ▷ Generator backward.
    $l_{\text{stu}} \leftarrow \mathcal{D}(\hat{\mathbf{x}}_{0,\text{stu}}), l_{\text{tea}} \leftarrow \mathcal{D}(\hat{\mathbf{x}}_{0,\text{tea}})$
    $\mathcal{L}_{\text{dis}} \leftarrow \texttt{cross\_entrophy}(l_{\text{stu}}, N + i) + \texttt{cross\_entrophy}(l_{\text{tea}}, i)$
    Update $\boldsymbol{\eta} \leftarrow \boldsymbol{\eta} - \frac{\partial}{\partial \boldsymbol{\eta}} \mathcal{L}_{\text{dis}}$ ▷ Discriminator backward.
  **until** converged

---

### A.5 INFERENCE SETTING

The prompts of generated samples are mainly drawn from MSCOCO (Lin et al., 2014), PartiPrompts (Esser et al., 2024), and open-source platforms. For SDv1.5 architecture, the generation resolutions contain 512x512, 512x768, and 768x512. For SDXL architecture, the generation resolutions contain 1024x1024, 1536x1024, and 1024x1536. The guidance (CFG) scale is set to 7 constantly, and we adopt a simple negative prompt *'worst quality,low quality,normal quality,lowres,watermark,nsfw'*. We use DPM-Solver++ (Lu et al., 2022a;b) scheduler, the number of inference steps is 25.

## B ADDITIONAL RESULTS

This section provides more results of generated samples under different tasks to demonstrate our approach's performance and flexibility.

### B.1 MAIN RESULTS

We display more results of our DMM for text-to-image generation with different styles in Fig. 10 and Fig. 11. It is worth noting that since some teacher models can generate images at multiple scales, our DMM can well inherit this ability even though it does not access the multi-scale data during training, as shown in Fig. 12. The results on SDXL are provided in Fig. 13. The results on SPLAM are provided in Fig. 16.

### B.2 RESULTS WITH PLUGINS

We display more results of our DMM equipped with various downstream plugins: ControlNet in Fig. 14 and LoRA in Fig. 15. It can be seen that our model maintains the power of these plugins while presenting different styles.

We show more results of DMM combined with Mixture-of-Diffusers in Fig 17, the panoramas that harmonize different styles.

### B.3 RESULTS OF STYLE MIXING

In this part, we illustrate the effectiveness of our proposed approach to style mixing. In Fig. 18, we interpolate the eight styles pairwisely with equal weights and show the grid of results. In Fig. 19, we perform interpolation on two styles and adjust the weights, delivering a smooth and natural transition between them.

## C LIMITATION AND FUTURE WORKS

As far as we know, we are the first to comprehensively analyze and reorganize the model merging task in the context of diffusion generative models. We are also the first to propose a baseline training method for diffusion model merging based on knowledge distillation. While training brings the advantage of performance improvements, it also requires more computing resources. Our DMM currently consumes about 32 GPU days for 100k training iterations to reach optimal convergence, which is relatively high. We hope for more observation and investigation in the scenario of diffusion model merging, and more efficient approaches can be explored.

A kangaroo wearing an orange hoodie and blue sunglasses standing on the grassin front of the Sydney Opera House

A brown teddy bear holding flowers in front of a grave

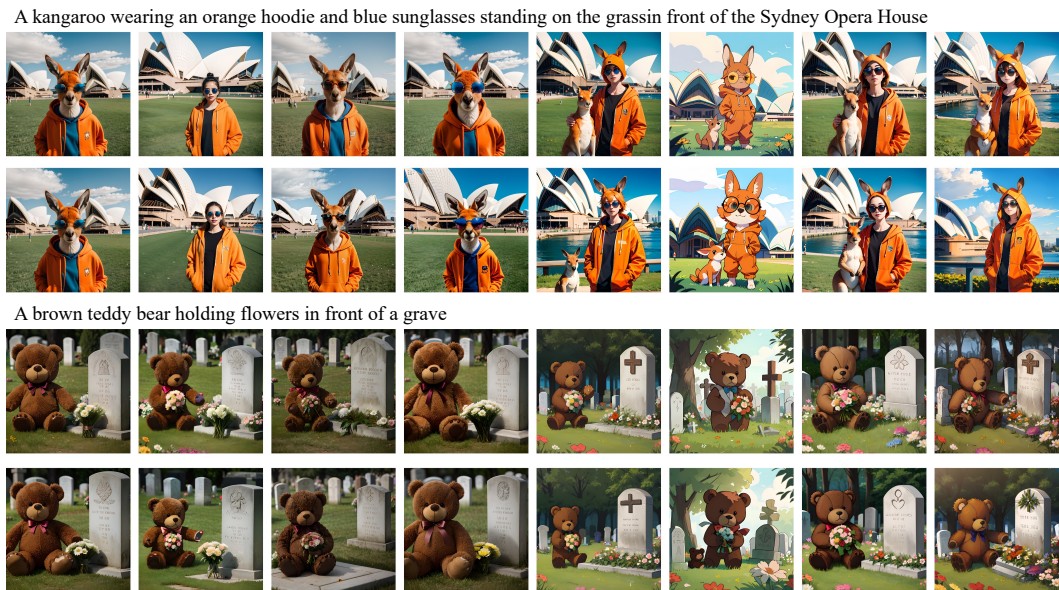

Figure 10: More text-to-image results of DMM compared with the teacher models. In each group, the first line is the results of DMM, and the second line is the results of teacher models.

1boy,handsome male,face,beard, beige shirt, white background

masterpiece,bestquality, 1girl, close up, colorful, cinematiclighting, bustshot, extremelydetailedCGunity8kwallpaper, redhair, solo,smile,intricateskirt,flyingpetal,Flowerymeadow sky, cloudy_sky, building, moonlight, moon, night, darktheme, light, fantasy,

World of ice and snow, Epic level scenery

a squirrel in a field

a boy and a penguin sitting on the moon

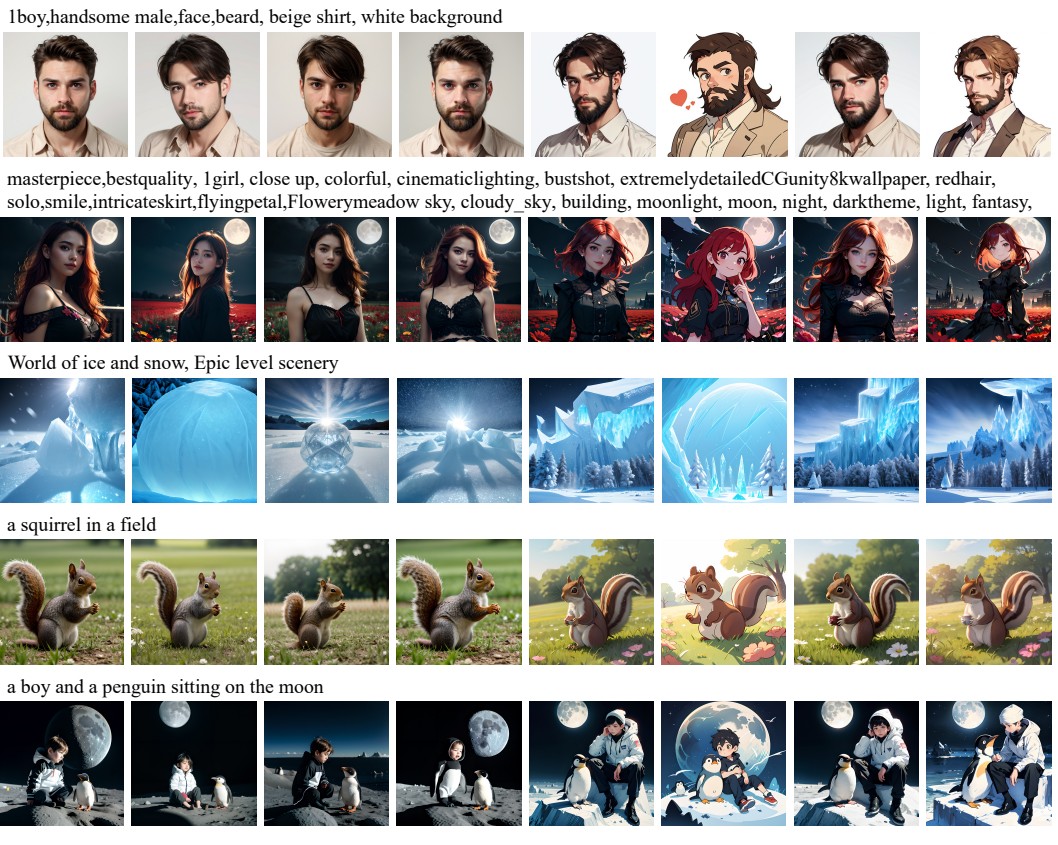

Figure 11: More text-to-image results of DMM.

masterpiece,best quality,igirl,solo,pinkeyes,stuffed bunny,white thighhighs,pinkhair,bangs,smile,looking at viewerlonghair,hair intakes,holding,dress,bareshoulders,closed mouth,virtual youtuber,offshoulder,holding stuffed toy,shoes,cowboyshot,ribbon,long sleeves,multicoloredhair,white jacket,pink ribbon,whitehair,bow,braid,white dress,hair betweeneyes, gradient background, absurdres,veryaesthetic,multicolored hair,amazingquality,detached sleeves

elf portrait,enchanting beauty,fantasy,ethereal glow,pointed ears,delicate facial features,long elegant hair,nature-themed attire,mystical ambiance,soft lighting,tranquil expression,harmonious with nature,subtle magical elements,serene,intricate jewelry,dreamlike quality,pastel colors,

1girl, black hair with bangs, white sweater, orange background

A truck that is sitting in the grass.

A young child using an electric sander on a long piece of wood.

Figure 12: More multi-scale text-to-image results of DMM.

mysterious silhouette woman with hat,by Minjae Lee,Carne Griffiths,Emily Kell,Steve McCurry,Geoffroy Thoorens,Aaron Horkey,Jordan Grimmer,Greg Rutkowski,amazing depth,double exposure,surreal,geometric patterns,intricately detailed, bokeh,perfect balanced,deep fine borders,artistic photorealism,smooth

a beautiful female model, In the studio, beautiful flowers, depth of field, brightly

masterpiece,best quality,1girl,fractal art,ink wash painting art style,realistic art style

1boy, young man black hair putting hair, male k-pop idol, lovingly looking camera, levi ackerman, medium portrait soft light, chop, beautiful model, oval face, vivid

backlit portrait of a girl, soft sunlight halo around her head, wearing a thick blue scarf and a black coat, straight face, light leaking from an overexposed window, neutral, relaxed expression, 25 years old, masterpiece

Figure 13: Text-to-image results of DMM on SDXL architecture.

a boy on the country road, best quality

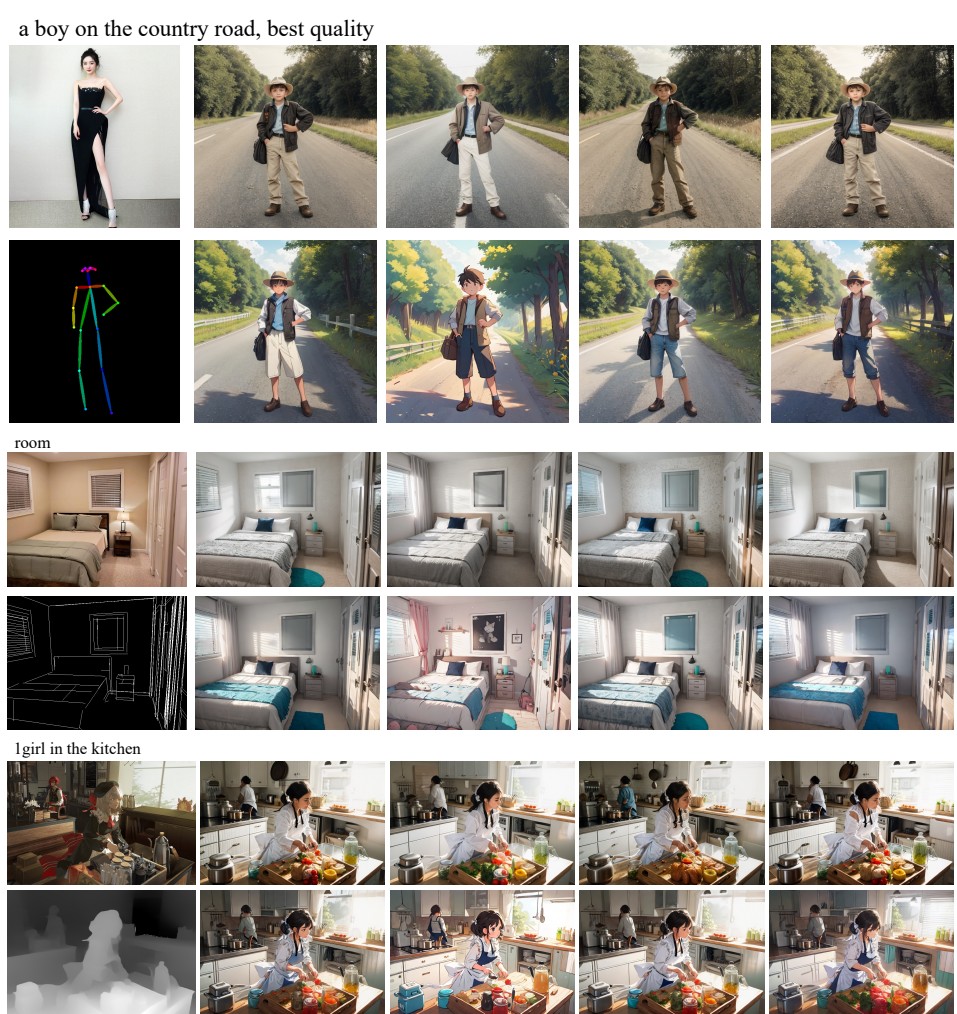

Figure 14: More results of DMM combined with ControlNet. We leverage the control conditions of OpenPose, MLSD, and Depth.

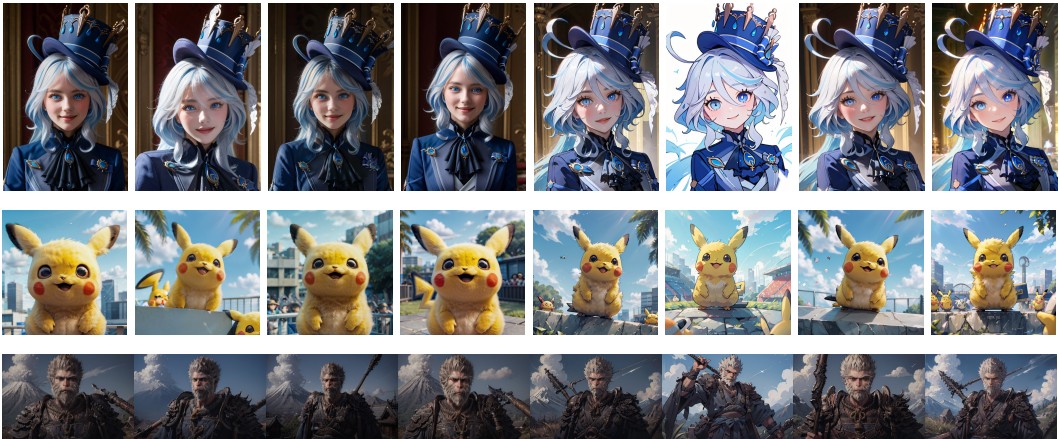

Figure 15: Results of DMM combined with different LoRAs. We use three open-source LoRAs respectively: Genshin Impact Furina, Pikachu, and Black Myth Wukong.

1girl, pink hair

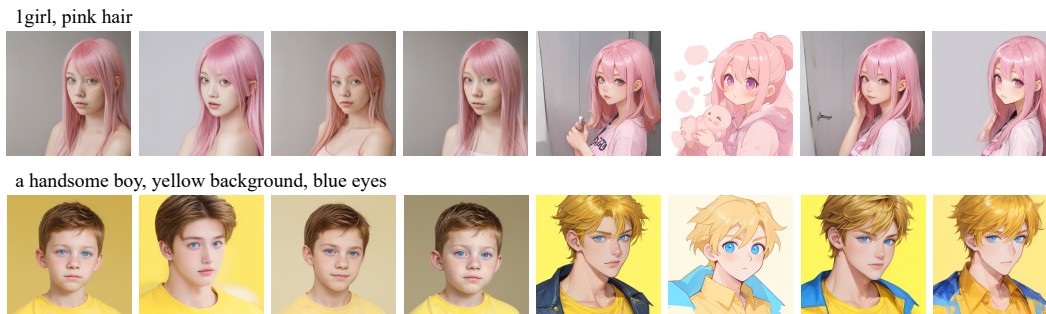

a handsome boy, yellow background, blue eyes

Figure 16: Results of DMM combined with SPLAM with only 4 inference steps.

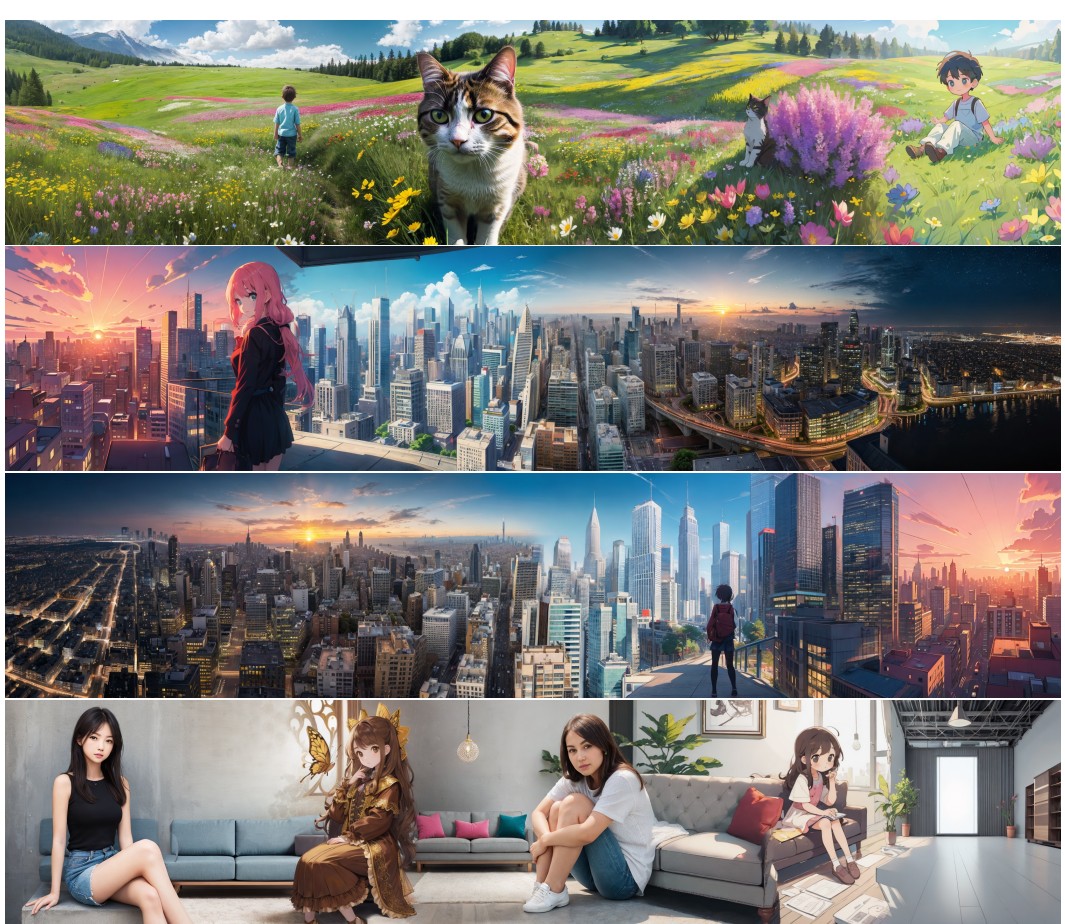

Figure 17: More results of DMM combined with Mixture-of-Diffusers.

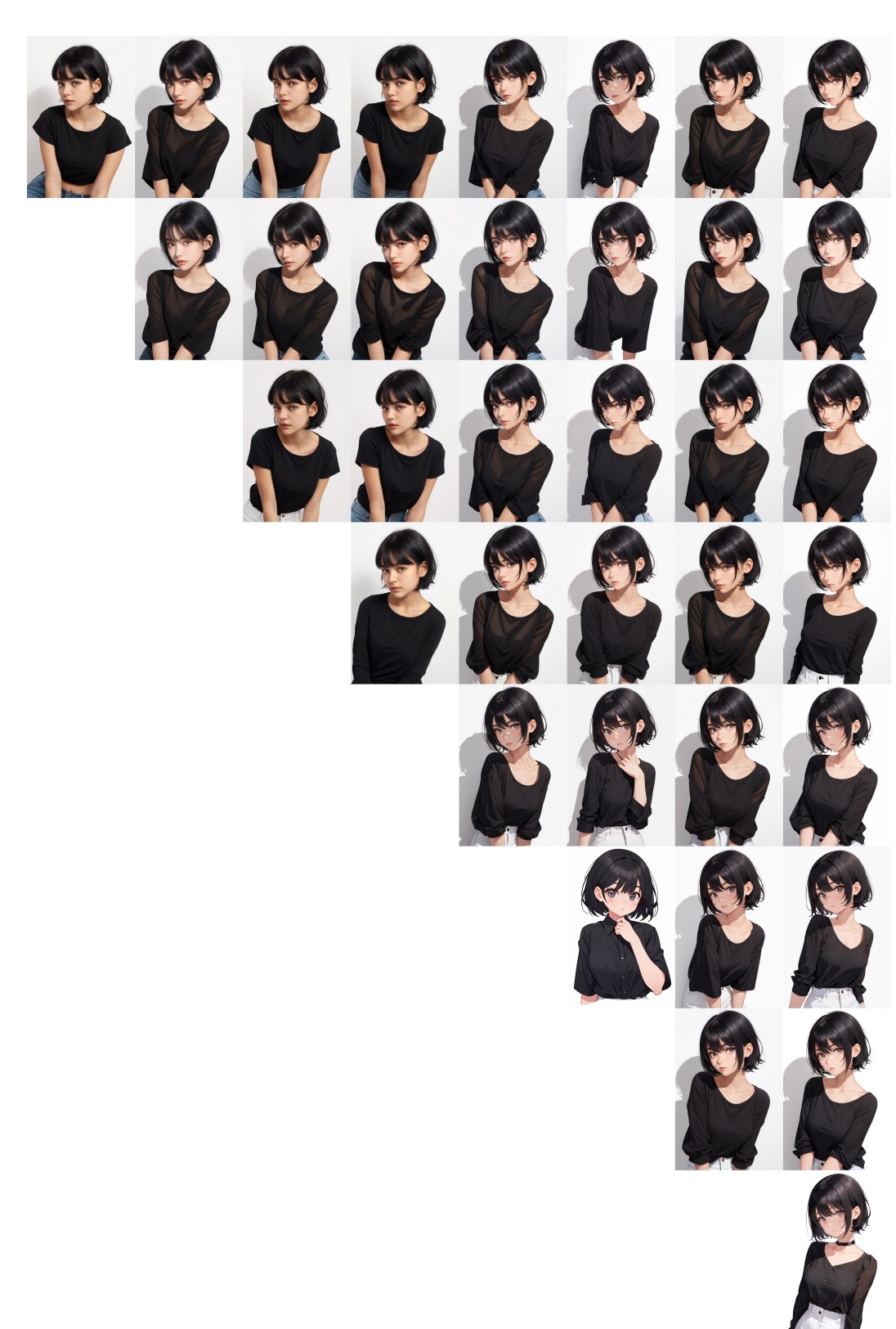

Figure 18: The result gird of pairwise interpolation. The interpolation weights are 0.5 and 0.5.

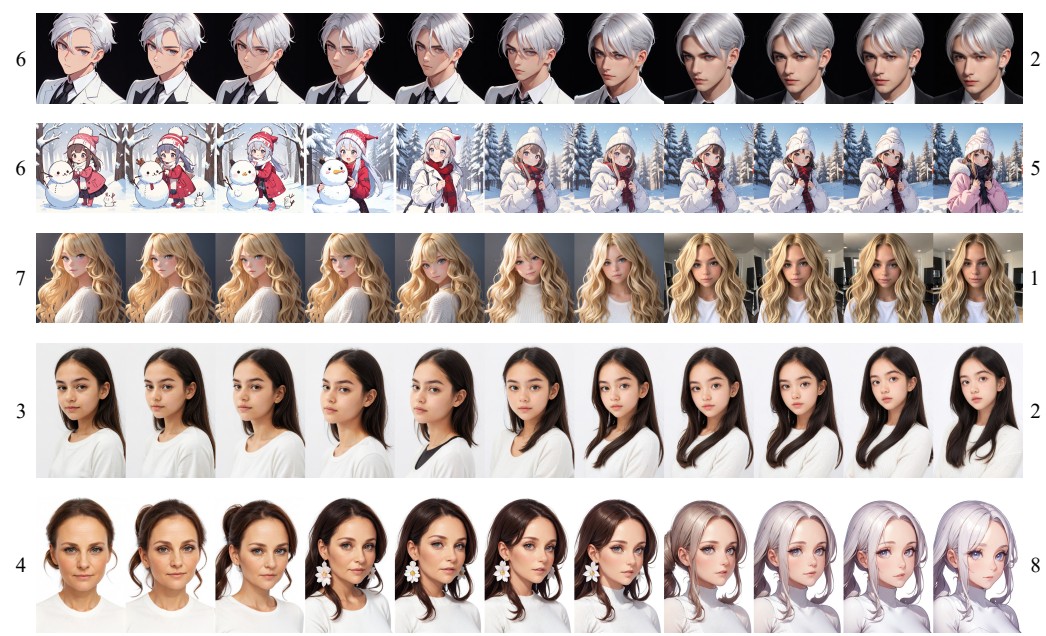

Figure 19: More results of interpolation between two styles. The number on the side is the model index. The weight list of one ingredient is [0.0, 0.1, 0.2, 0.3, 0.4, 0.5, 0.6, 0.7, 0.8, 0.9, 1.0].

