# DMM: Building a Versatile Image Generation Model via Distillation-based Model Merging

## Rebuttal

### 0.1 Responses to Reviewer Za1P

**Q.1 The selection of styles is too weak.**    To further demonstrate the generalizability of our approach, we conduct experiments on a wider variety of distinct models, including Chinese ink painting, pixel art, business illustration, etc. As shown in Sec.0.3, our DMM still works well on these challenging models.

**Q.2 Demonstration of issues being addressed.**    To concretely demonstrate that directly merging models of various styles will cause conflict and style confusion, we conduct such experiments. With the same setting of model candidates, namely the eight models with different styles, we apply weighted merging directly on their parameters and use the merged model to synthesize images. As shown in Fig 1, the output of this model exhibits an ambiguous hybrid style, rather than being purely realistic or entirely anime-inspired. Even with additional corresponding keywords (e.g. *'realistic'*, *'animation'*), it still struggles to accurately distinguish between different styles and cannot produce satisfactory results, compared to our results in Figure 7 in the main paper. This result is not what we expected and we cannot effectively control the generation. This phenomenon can also be used to explain why prompt engineering cannot achieve the same effects as our merging approach, since handcraft prompt tuning operates at a relatively vague level, while our method can provide precise suggestions and access expert model knowledge.

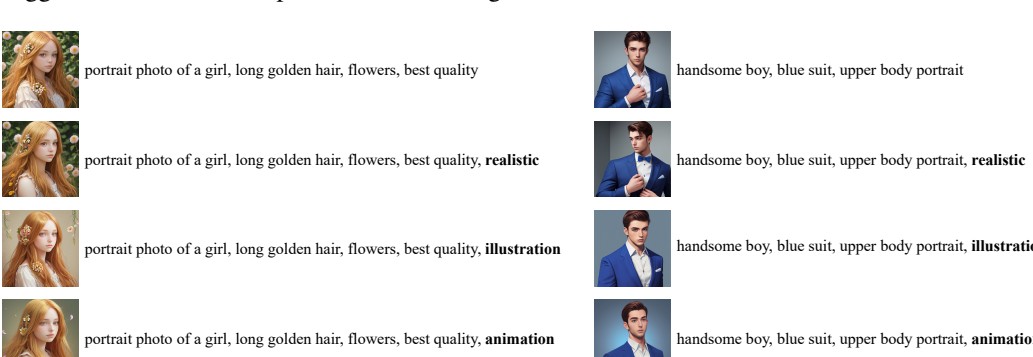

Figure 1: Results of the simply merged model (weighted merging). Parameter merging of different or even opposite styles causes serious conflicts and thus leads to style confusion.

**Q.3 Baseline results in Table 1.**    We use the initialized student model, namely SDv1.5 as the baseline results. Specifically, for the FID calculation of each style, we use additional style-related keywords in the text prompt to modulate different styles.

**Q.4 Continual learning unrelated to the core problem of model merging.**    That is not the case. The sustainability of our method is crucial for the application of model merging and should be paid attention to. If our approach is a disposable solution, its application flexibility would be greatly limited. Users often encounter situations where they want to incrementally merge a recently developed model into the current checkpoint. If they have to train from scratch, it will bring inconveniences and resource waste, while our continual learning strategy alleviates the problem.

| Methods | FIDt↓ |
|---|---|
| Baseline | 195.22 |
| + Score Distillation | 80.69 |
| + Feature Imitation | 79.27 |
| + Multi-Class Adversarial | 78.38 |
| + Synthesized Finetune | **77.51** |
| Teacher Reference | 74.91 |

Table 1: **Ablation Results**. The second to fourth lines are our proposed three types of losses. The fifth line is the fine-tuning stage with synthesized data.

**Q.5 Visual results for ablation configurations.** We provided visual results of different ablation settings to verify the improvement in visual quality brought about by the reduction in FID. It can be seen that, as the FID decreases, the images exhibit more refined optimizations, such as improved textures and lighting, indicating a closer alignment with the target style.

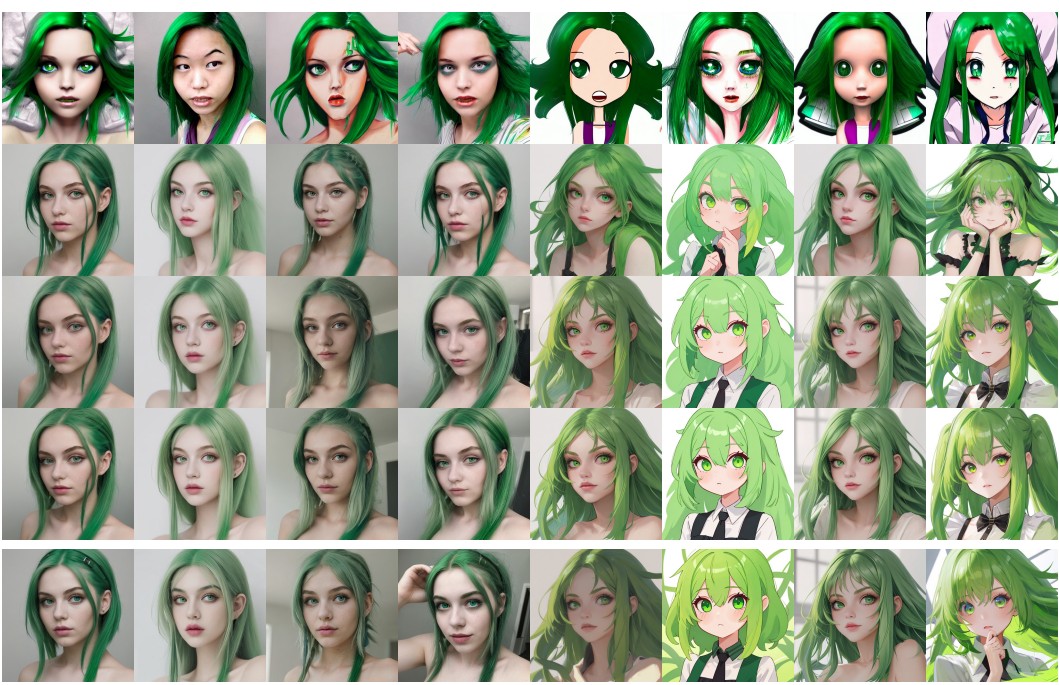

Figure 2: Visual results for ablation configurations. The first line is the baseline results with FID 195.22. The FID results for the second to fourth lines are 80.69, 79.27, and 77.51, respectively. The last line is the results of teacher models.

**Q.6 More insights from Table 2 results in continual learning.** First, considering experiments #1 and #2, we can observe that for the first four FIDs, the results of #2 are slightly worse than those of #1. This potentially hints at a scaling relationship between performance and the number of models, and it is understandable, as the increase in the number of models would challenge the overall capacity and performance of the models. Fortunately, this performance degradation is still quite manageable within a larger scope, at around 1%. Second, considering the #3 and #4, freely fine-tuning the last four models in #3 shows a slight advantage compared to #4, but introduces unacceptable bursts in the first four FIDs. It is desirable to employ our proposed regularization method to achieve balanced results. Besides, these insights will also inspire us to analyze the scaling laws regarding the number of models and the performance.

**Q.7 Clarification of results compared to weighted merging in Table 3.** In the first place, Table 3 provides one perspective on the capabilities of our model, aiming to demonstrate that our approach can cover the effects of original common practices by performing mix augmentation within simi-

lar domains. In addition, our method offers more advantages in terms of flexibility compared to weighted merging, as claimed in the paper.

As for the determination of the weights, the optimal weights should be searched manually for both methods. However, the original method is disposable since the parameters cannot be adjusted once the weight-merged model file is dumped. In contrast, our method allows for free weight control during inference time through input arguments based on current requirements, making it much more flexible.

**Q.8 Requires access to the fine-tuned models.** The goal of the model merging task is to integrate multiple fine-tuned models, which naturally require access to them. This is the same in many other task fields, such as the merging of LLMs, detection models, etc. We are truly addressing practical issues in the T2I community: users want to utilize these ready-made expert models more efficiently and flexibly.

**Q.9 Typos for the term DDPM and GAN.** We apologize sincerely for the typos, and thoroughly checked and rectified them. Thanks for pointing out these typos.

**Q.10 Nature of the style prompts.** We have provided the implemented details of style prompts in the Appendices. Sorry for not making it clear, we will place this important information in the main text rather than in an appendix. Specifically, we first align the embeddings with an MLP and then add it to the timestep embedding, which is lightweight enough for plug-and-play purposes. The shape of style prompts is $(N, 1280)$, where $N$ is the number of models. The MLP contains two $(1280, 1280)$ linear projection. That is all the additional trainable parameters.

0.2 RESPONSES TO REVIEWER SQM4

**Q.11 Concern about application.** There is indeed a demand for model merging techniques within the community; in fact, this technology is needed in our industrial products. Even though there are LoRAs available to achieve lightweight models, achieving satisfactory strong performance still requires full parameter fine-tuning and most top powerful models are still in the form of base models in the communities. In situations, we exactly should use the satisfying models and only have access to the base model (maybe the LoRAs are merged into the UNet), our approach is proposed to address this issue. Besides, the frequently used weighted merging is also trying to achieve the target but it is not flexible enough, so the application of our approach is desirable among the community.

**Q.12 Necessity of model merging.** As mentioned above, the situation our method aims to address is the desire to flexibly and efficiently utilize existing expert models in the community, as these expert models are highly valuable due to being fine-tuned on high-quality datasets and can accurately satisfy our application requirements. The SDXL can cover many styles but it is too generalized and coarse. Compared to specific expert models, prompt tuning alone is very inefficient and falls far short of meeting application requirements. As shown in Figure 3, we try to control the style by adding keywords on SDXL, but the results are far from reaching the performance of our approach and other advantages of our DMM like style mixing are completely impossible.

portrait photo of a girl, long golden hair, flowers, best quality

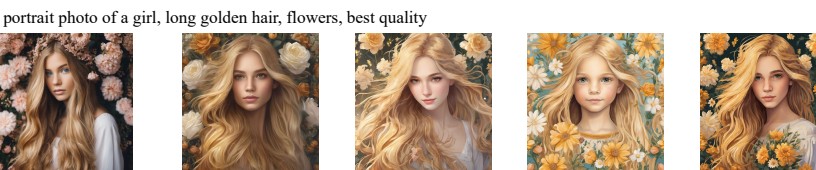

+realistic          +animation          +illustration          +pixel art

Figure 3: Results of vanilla SDXL. The style-related keywords are added to the text prompt to modulate the style.

**Q.13 Clarification for Table 3.** Table 3 provides a perspective on the capabilities of our model, aiming to demonstrate that our approach can achieve results comparable to the original common practices by performing mix augmentation within similar domains. More investment can bring

more advantages. Our approach can gather knowledge from many distinct models and apply flexible adjustment at inference time, while weighted merging cannot be adjusted after fixing the weights.

**Q.14 Analysis of the effect of the number of teacher models.** Sincerely thanks for your key suggestion for our approach principles as this determines the scalability of our DMM. We conducted experiments with three different settings for the number of teacher models, as shown in Table 2. It should be acknowledged that the FID for each target slightly increases with the number of teacher models. Fortunately, this performance degradation is still quite manageable within a larger scope, at around 1% every addition of 10 models. This guarantees that our method can be effectively scaled to tens of teacher models.

| #Model | 1 | 2 | 3 | 4 | 5 | 6 | 7 | 8 | 9 | 10 | 11 | 12 | 13 | 14 | 15 | 16 |
|---|---|---|---|---|---|---|---|---|---|---|---|---|---|---|---|---|
| 4 | 9.5 | 9.8 | 9.5 | 9.7 | - | - | - | - | - | - | - | - | - | - | - | - |
| 8 | 9.6 | 9.8 | 9.6 | 9.9 | 9.8 | 9.0 | 9.8 | 10.0 | - | - | - | - | - | - | - | - |
| 16 | 9.6 | 9.9 | 9.7 | 9.9 | 9.9 | 9.1 | 9.8 | 10.1 | 9.9 | 9.8 | 9.8 | 9.9 | 9.5 | 10.0 | 9.9 | 9.8 |

Table 2: Results with different numbers of teacher models.

**Q.15 What are the base models in Figure 3?** The information on all the teacher models is provided in the Appendices. As stated in the manuscript, we merge the first four styles, namely the base models are the first four models in Table 4.

**Q.16 Concern about style overlap.** This will not cause problems. Similar styles will only simplify the training task, and even models with similar styles will have subtle differences. Our approach is capable of learning the specific knowledge and faithfully reproducing each teacher model.

Additionally, we conduct experiments on more special-style models as demonstrated in Sec. 0.3, and our approach still works well.

**Q.17 Applicability of merging LoRAs.** Yes, it is feasible to apply our method for merging LoRAs. In fact, in the above experiment of unique style, the teacher models are in the form of LoRAs, and we successfully merge them.

0.3 RESPONSES TO REVIEWER V6ZA

**Q.18 Exploration of more unique styles.** We conducted experiments on four unique challenging styles, including the instances you provided. The detailed information is shown in Table 3 below. The main results are in Figure 4 and the mixing results are in Figure 5, which reveals that our approach can generalize to a broader range of distinct styles. It is worth noting that these models are in the form of LoRA, which also validates the feasibility of our approach in merging LoRAs.

| No. | Model Name | Style | Source |
|---|---|---|---|
| 1 | CJIllustration | Business flat illustrations | https://www.liblib.art/modelinfo/ |
| 2 | MoXin | Traditional Chinese ink painting | https://civitai.com/models/12597 |
| 3 | MPixel | Pixel art | https://civitai.com/models/44960 |
| 4 | SCHH | Natural scenery | https://www.liblib.art/modelinfo |

Table 3: The information of all models with rare and unique styles.

**Q.19 Incremental learning with enhanced setting.** We conduct the incremental experiment setting the initial task with models of different styles, namely the models with index $[1, 2, 5, 8]$. The results are shown in Table 4, with teacher reference results provided. It can be seen that the choice of the teacher models in the initial stage has little impact on the performance of our method.

**Q.20 Initialization of the student model.** For a simple and general purpose, we initialize the student using SD1.5 weights. We will add the information in the manuscript.

tree,day,water,blue sky,forest,rock

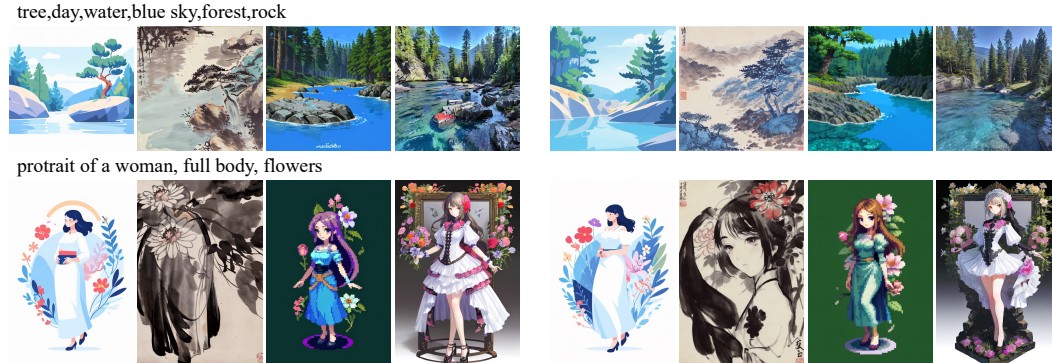

protrait of a woman, full body, flowers

Figure 4: Visual results with different style selections. In each group, the left part is the teacher models' results, the our model's results.

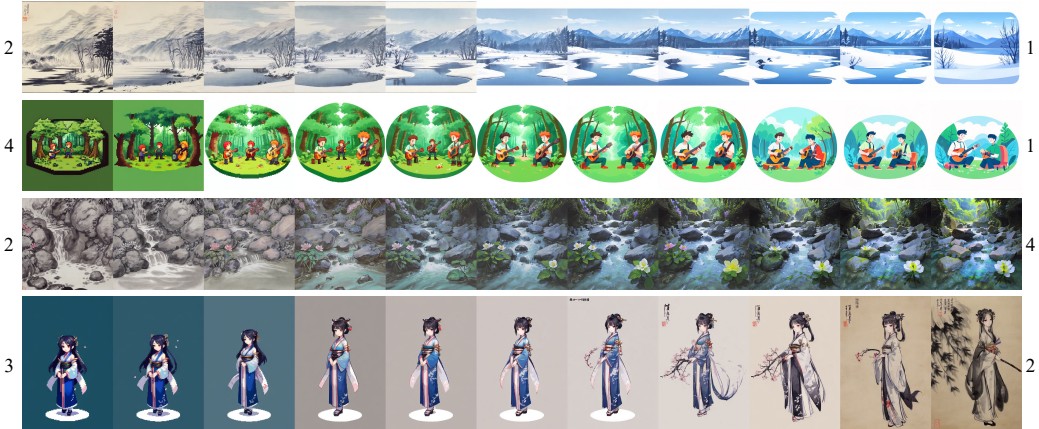

Figure 5: Results of interpolation between two styles. The number on the side is the model index.

**Q.21 Outcome with no style prompts for input.** We conducted this experiment and found that when no style prompt is provided (i.e. the time embeddings are added with zeros), the model's output is more like an average of the contained styles. We further validate this point by calculating the norm of each style embedding, obtaining that the norm of each 1280-dimensional vector is approximately 36, while the norm of the average of them is a lot smaller (about 12).

Actually, we have considered regularizing the model prediction with null-style conditions, where we can leverage the vanilla SD1.5 for supervision. However, according to the goal of the practical task, which is to learn from the knowledge of different expert models, this operation is unnecessary. Frankly, as a foundation model, the images generated by SD1.5 are not aesthetically pleasing; it only provides nice initialization for fine-tuning. Therefore, in practice, we are more focused on utilizing these fine-tuned upgraded models. We prefer to allow the zero-embedding input to become an adaptive result in embedding space under several anchor points, where the results still exhibit the knowledge of the experts rather than the relatively unappealing output of SD1.5.

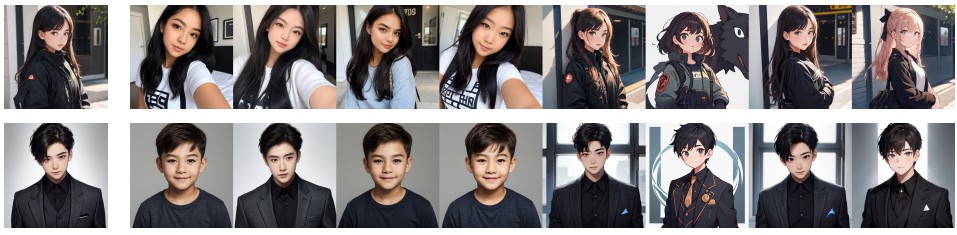

Figure 6: The left images are the results with no style prompts selected. The right images are the results with different style prompts.

| # | Stage | #Model | 1 | 2 | 3 | 4 | 5 | 6 | 7 | 8 |
|---|-------|--------|---|---|---|---|---|---|---|---|
| 0 | Teacher | 8 | 9.5 | 9.5 | 9.2 | 9.5 | 9.5 | 8.2 | 9.7 | 9.7 |
| 1 | Train | 8 | 9.6 | 9.8 | 9.6 | 9.9 | 9.8 | 9.0 | 9.8 | 10.0 |
| 2 | Train | 4 | 9.5 | 9.8 | 9.5 | 9.7 | - | - | - | - |
| 3 | Fine-tune | 4 + 4 | 9.7 | 9.8 | 9.6 | 9.8 | 9.9 | 9.0 | 9.8 | 10.0 |
| 4 | Train | 4 | 9.6 | 9.9 | - | - | 9.8 | - | - | 9.9 |
| 5 | Fine-tune | 4 + 4 | 9.7 | 10.0 | 9.6 | 9.8 | 9.8 | 9.0 | 9.8 | 10.1 |

Table 4: **Incremental Learning**. #0 denotes the reference results of teacher models. #3/5 is fine-tuned from the checkpoint of #2/4 respectively. #2 and #4 select different subsets of styles.

**Q.22 Explanation for consistency not only in style but also in details.** To explain the phenomenon, it is because the foundational target of our task is to reproduce every teacher model faithfully. Thus, we resort to leverage knowledge distillation to achieve this. Style is just one of the more obvious and intuitive aspects of model **knowledge**, and we primarily use the term "style" in the paper to describe the characteristics of distinct models. In fact, during the distillation process, our approach can distill a wider range of both high-level and low-level knowledge from models such as layout and pose preference, through our supervision of scores, features, and adversarial objectives. We consider this point to be important and a primary principle of the task, because in some practical applications, we exactly need the effects of these expert models, which previous merging methods have been unable to achieve. The 'style prompts' at the timestep embedding level play the role of switching among the modes and it is enough to work well.

## 0.4 RESPONSES TO REVIEWER JLMO

**Q.23 Clarification of the learnable embeddings.** We have provided the implemented details of style prompts in the Appendices. Sorry for not making it clear, we will place this important information in the main text rather than in an appendix. Specifically, we first align the embeddings with an MLP and then add it to the timestep embedding, which is lightweight enough for plug-and-play purposes. The shape of style prompts is $(N, 1280)$, where $N$ is the number of models. In this way, we can modulate the model in both the training and inference phases and make sure it learns MoE-like ability.

**Q.24 Exploration on merging of multiple control conditions.** In principle, the goal of merging multiple control conditions is achievable because our approach is essentially to provide signals to switch between different modes. We conduct experiments on merging three types of ControlNet: canny, depth, and pose. Besides, the merging of ControlNets and UNets is orthogonal and can be combined compatibly. We show the results in Figure 7, and it is worth mentioning that this versatile effect is achieved with only one single UNet and only one ControlNet, demonstrating the potential of broader applicability tasks of our approach.

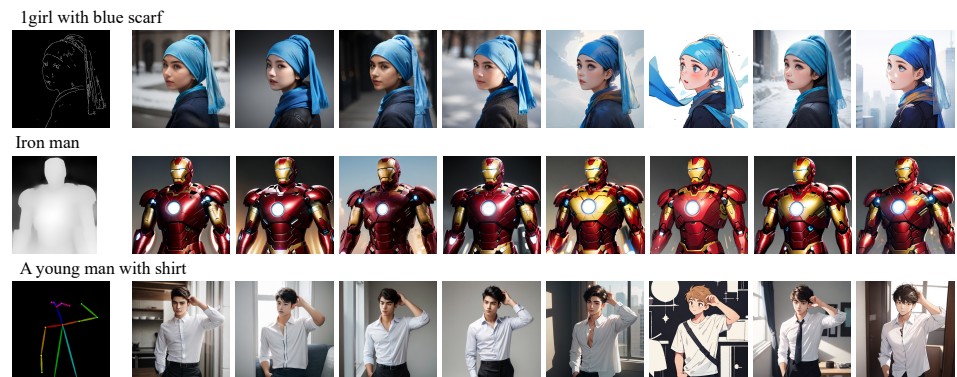

Figure 7: Visual results of the combination of our DMM-UNet and DMM-ControlNet. The multiple styles and multiple controls are achieved with only a single model.