# OpenReview forum: "DMM: Building a Versatile Image Generation Model via Distillation-Based Model Merging"
_ICLR.cc/2025/Conference — Submitted to ICLR 2025_

### Official Review · Reviewer_JLMo · 2024-10-30

**Soundness:** 3
**Presentation:** 3
**Contribution:** 2
**Rating:** 5
**Confidence:** 4

**Summary:**

The authors propose that the ability of expert models to contain many community models should be refined, while avoiding the proliferation of downstream models.This is a good insight. The authors propose methods based on score distillation, feature constraints, and adversarial learning to address the challenges presented. Unfortunately, the authors introduce a more ambitious vision in INTRODUCTION, but actually generalize only style learning.

**Strengths:**

1 The ideas presented by the author are reasonable and interesting.

2 The ability to distill a strong teacher model to learn more student models is something the community needs and should be tapped into.

**Weaknesses:**

1 Learnable Embeddings didn't make it clear? How size it is and how to make sure it learns moe-like abilities.

2 The introductory section is suspiciously exaggerated and actually lands in the style section. I think it's more important to explore how it can learn the capabilities of more models, such as animation, and the merging of multiple control conditions is what the community needs.

3 How can we quantify the ability of a distilled model to learn from multiple models without degrading the original model? This is what would like to see.

**Questions:**

1 Learnable Embeddings didn't make it clear? How size it is and how to make sure it learns moe-like abilities. Is it just the style capabilities available, I think the community needs to migrate more than just style.

2. If it's just the style migration ability, the author should compare it to articles like IP-Adapter,Instantstyle,instanstyle plus,styleshot,CSGO.

3. I believe the author's motivation is reasonable, but there should be programs with broader applicability for more tasks to consider.

4. see weaknesses.

---

> ### Author Response · Authors · 2024-11-22
>
> **Thanks for the valuable feedback**
>
> Thank you for taking the time to provide your valuable feedback.
> For the visual demonstration, we provide image results in the **Supplementary Material PDF (cause we can not directly submit figs on openreview).**
>
> **Q.1 Clarification of the learnable embeddings.**
>
> We have provided the implemented details of style prompts in the Appendices.
> Sorry for not making it clear, we will place this important information in the main text rather than in an appendix.
> Specifically, we first align the embeddings with an MLP and then add it to the timestep embedding, which is lightweight enough for plug-and-play purposes.
> The shape of style prompts is $(N, 1280)$, where $N$ is the number of models.
> In this way, we can modulate the model in both the training and inference phases and make sure it learns MoE-like ability.
>
> **Q.2 Exploration on merging of multiple control conditions.**
>
> Thank you for posting such an insightful point of view.
> In principle, the goal of merging multiple control conditions is achievable because our approach is essentially to provide signals to switch between different modes.
> We conduct experiments on merging three types of ControlNet: canny, depth, and pose.
>
> Besides, the merging of ControlNets and UNets is orthogonal and can be combined compatibly.
> We show the results in **Figure 7 in the PDF**, and it is worth mentioning that this versatile effect is achieved with only one single DMM-UNet and only one DMM-ControlNet, demonstrating the potential of broader applicability tasks of our approach.
>
> **Q.3 Compared to other style migration methods**
>
> The motivation of our DMM and these methods are different. Our approach aims to resemble and reorganize the knowledge from different teacher models. As demonstrated above, our DMM is **not just for the style migration ability**, though style information may be an explicit component of so-called model *knowledge*.
> Our approach can be applied to more programs, such as merging multiple control conditions.

---

> > ### Comment · Reviewer_JLMo · 2024-11-27
> > **Response**
> >
> > Thank you for your response. I still remain skeptical. The authors provide a good perspective, but the results are not satisfactory, especially whether the learnable embedding in 1280 dimensions can be used as a MoE capability for multiple models.

---

> ### Author Response · Authors · 2024-11-27
>
> Thank you for your response.
>
> For the mechanism of learnable embeddings, we exactly follow the good practice of SDXL's micro-condition, which also encodes the *original image size, target image size, and cropping parameters* into 1280-dimension embeddings and add them to the timestep embedding, and have been extensively validated as effective.
> From our visual results, it is evident that such a simple approach is sufficient to guide the model in learning different target knowledge.
> It is distinct that our model can clearly differentiate between various experts and faithfully reproduce the generation of the original teacher model.
> Additionally, our method generalizes beyond just style information, as discussed in **Q.5 posed by Review V6za regarding details preference**, as well as in **the multi-control merging you suggest**.
> With the support of both quantitative and qualitative performance, we believe that this demonstrates that our 1280-dimension embedding provides a simple yet effective merging method instead.
>
> We honestly show that the learnable embedding can be used to derive our current moe-like results and verify its potential, by serving as a flexible switch.
> We hope that this can relieve your concern or you can identify your concrete dissatisfaction regarding the demonstration presented in our paper.

---

### Official Review · Reviewer_V6za · 2024-10-31

**Soundness:** 3
**Presentation:** 3
**Contribution:** 2
**Rating:** 5
**Confidence:** 4

**Summary:**

This paper proposes DMM, a novel approach for compressing multiple models into a single versatile text-to-image (T2I) model. DMM integrates the styles of several models into a unified student model through knowledge distillation and continual learning. A style codebook is designed to conveniently control which teacher model’s style is followed. The authors introduce FIDt, a evaluation metric, on which DMM achieves excellent performance. This method offers a new perspective on model merging tasks.

**Strengths:**

1. The paper presents an approach to compress multiple models into a single versatile text-to-image (T2I) model. The originality lies in proposing a solution that integrates the strengths of various T2I models. Based on distillation, this method combines multiple T2I models into one. To allow for the specification of which model to utilize, the authors have designed a codebook that enables convenient control.
2. This work has promising application potential and practical value. Current community models have their unique strengths, and this method offers a valuable approach for optimizing and integrating diverse community models.

**Weaknesses:**

1. Due to the lack of direct access to the original training data for each teacher model, the authors suggest treating this optimization process as a conventional regression task, using a generalized dataset for training. This approach would be effective if the dataset can adequately capture the core characteristics of the teacher models. However, when attempting to incorporate unique style teacher models, this method may be less effective. The study would be more convincing if the authors explored more distinctive styles beyond the common realistic and cartoon styles.
2. In incremental learning , the paper sets the initial task as four realistic-style models and a new task as four cartoon-style models. This setup may not adequately demonstrate the method's ability to resist catastrophic forgetting in incremental learning. Due to the distinct difference between the initial and new styles ,  it is easy for the model to distinguish between initial and new tasks . This may not effectively demonstrate the method’s ability to handle confusion. To provide more convincing results, the author maybe consider setting the initial task with models of different styles and introducing multiple new tasks, each introducing models of a different style.
3. The paper primarily presents results on a limited set of styles. The authors list eight community models in SDv1.5; however, these largely represent only two broad categories: Realistic and Anime & Illustration & 3D Cartoon. This limited scope makes it challenging to assess the effectiveness of style mixing. To better demonstrate the method's capability in style mixing, it is recommended to experiment with a wider variety of distinct style combinations, such as MoXin, M_Pixel, Miniature World Style and other community models.

**Questions:**

1, Was the model initialized using the SD1.5 weights, or was it initialized randomly? I did not find a detailed explanation regarding this aspect in the manuscript.
2, The authors demonstrate the selection of a single “style prompt” from N options or a combination of multiple “style prompts”. What would be the outcome if none of these “style prompts” were selected? Would the result resemble the original SD1.5 output?
3, I use fixed seed and prompt to generate images across different models. Sometimes the results differ so much that they appear unrelated, while other times they look like variations of the same image. This phenomenon is also evident in the author's Figure 6. I initially expected the author's model to maintain stylistic consistency with the teacher model, but was pleasantly surprised to see that it also retains similarity in details, as shown in Figure 6. I’m curious about what contributes to this effect. Just the ‘style prompt’ at the timestep embedding level?

---

> ### Author Response · Authors · 2024-11-22
>
> **Thanks for the valuable feedback**
>
> Thank you for taking the time to provide your valuable feedback.
> For the visual demonstration, we provide image results in the **Supplementary Material PDF (cause we can not directly submit figs on openreview).**
>
> **Q.1 Exploration of more unique styles.**
>
> Sincerely thanks for your insightful suggestion and for providing more interesting examples of our approach.
> We conducted experiments on four unique challenging styles, including the instances you provided.
> The detailed information is shown in the table below.
> The main results are in **Figure 4** and the mixing results are **in Figure 5 in the PDF**, which reveals that our approach can generalize to a broader range of distinct styles.
> It is worth noting that these models are in the form of LoRA, which also validates the feasibility of our approach in merging LoRAs.
>
> |No.|Model Name|Style|
> |-|-|-|
> |1|CJ_Illustration|Business flat illustration|
> |2|MoXin|Traditional Chinese ink painting|
> |3|MPixel|Pixel art|
> |4|SCHH|Natural scenes|
>
> **Q.2 Incremental learning with enhanced setting.**
>
> We conduct the incremental experiment setting the initial task with models of different styles, namely the models with index $[1, 2, 5, 8]$.
> The results are shown in the table below, which is also shown in **Table 4 in the PDF**.
> It can be seen that **the choice of the teacher models in the initial stage has little impact on the performance of our method**.
>
> |#|Stage|#Model|1|2|3|4|5|6|7|8|
> |-|-|-|-|-|-|-|-|-|-|-|
> |0|Teacher|8|9.5| 9.5| 9.2| 9.5| 9.5| 8.2| 9.7| 9.7|
> |1|Train|8|9.6| 9.8| 9.6| 9.9| 9.8| 9.0| 9.8| 10.0|
> |2|Train|4|9.5| 9.8| 9.5| 9.7| - | - | - | - |
> |3|Fine-tune|4+4|9.7| 9.8| 9.6| 9.8| 9.9| 9.0| 9.8| 10.0|
> |4|Train|4|9.6| 9.9| -| -| 9.8| -| -| 9.9|
> |5|Fine-tune|4+4|9.7| 10.0| 9.6| 9.8| 9.8| 9.0| 9.8| 10.1|
>
> The experiment #3 is fine-tuned from #2, and the experiment #5 is fine-tuned from #4.
>
> **Q.3 Initialization of the student model.**
>
> For a simple and general purpose, we initialize the student using **SD1.5 weights**.
> We will add the information in the manuscript.
>
> **Q.4 Outcome with no style prompts for input.**
>
> We conducted this experiment and found that when no style prompt is provided (i.e. the time embeddings are added with zeros), the model’s output is more like **an average of the contained styles**.
> We further validate this point by calculating the norm of each style embedding, obtaining that the norm of each 1280-dimensional style vector is approximately 36, while the norm of the average of them is a lot smaller (about 12).
>
> Actually, we have considered regularizing the model prediction with null-style conditions, where we can leverage the vanilla SD1.5 for supervision.
> However, according to the goal of the practical task, which is to learn from the knowledge of different expert models, this operation is unnecessary.
> Frankly, as a foundation model, the images generated by SD1.5 are not aesthetically pleasing; it only provides nice initialization for fine-tuning.
> Therefore, in practice, we are more focused on utilizing these fine-tuned upgraded models and we do not have to let the null-style outputs resemble the SD1.5 outptus.
> We prefer to allow the zero-embedding input to become an adaptive result in embedding space under several anchor points, where the results still exhibit the knowledge of the experts rather than the relatively unappealing output of SD1.5.
>
> **Q.5 Explanation for consistency not only in style but also in details.**
>
> Thank you for posting such an insightful point of view.
> To explain the phenomenon, it is because the foundational target of our task is to reproduce every teacher model faithfully.
> Thus, we resort to leverage knowledge distillation to achieve this.
> Style is just one of the more obvious and intuitive aspects of model **knowledge**, and we primarily use the term "style" in the paper to describe the characteristics of distinct models.
> In fact, during the distillation process, our approach can distill a wider range of both high-level and low-level information from models such as layout and pose preference, through our supervision of scores, features, and adversarial objectives.
> We consider this point to be important and a primary principle of the task, because in some practical applications, we exactly need the effects of these expert models, which previous merging methods have been unable to achieve.
> The `style prompts' at the timestep embedding level play the role of switching among the modes and it is enough to work well.

---

> > ### Author Response · Authors · 2024-11-29
> >
> > Dear reviewer, as the author's posting message ends on December 2, we want to ensure that all your questions have been thoroughly addressed. Your feedback is instrumental to us, and we would be grateful if you could spare a moment to provide a final rating and share your thoughts. Your input will greatly inform our future improvements.

---

### Official Review · Reviewer_SqM4 · 2024-11-02

**Soundness:** 4
**Presentation:** 3
**Contribution:** 2
**Rating:** 5
**Confidence:** 5

**Summary:**

This paper proposes a model merging paradigm based on score distillation to merge several models into a single versatile model. They present a  distributed training framework to implement the score-distillation, where multiple teacher models are distilled into a single student model (share the same UNet architecture as teacher models) using a MSE loss, a learnable embedding is used to distinguish each teacher model. Meanwhile, they use feature supervision (also via a MSE loss) to facilitate knowledge transfer. Finally, they incorporate an additional GAN objective to distinguish different style distributions.

**Strengths:**

- This paper is well-written and easy to follow.
- The proposed pipeline is reasonable, using a learnable embedding as style prompt is interesting.

**Weaknesses:**

- My major concern is about its application. The goal of merging multiple base models (teachers) is unclear. This paper argues that specialized model need redundancy and huge storage cost, but most of style models are in LoRA format (may be merged into UNet and publish as a base model), which is already a light-weighting model and can be used as a plugin.
- For SDXL, it can already cover many styles via prompting, and selected teacher models are usually finetuned to a specific style. In what case we need re-merge these finetuned models into a versatile model?
- As shown in Table 3, the performance gain with weighted merging is minor, while the later is much easier.
- Only eight teacher models are used in experiment. It would be great to analyze the effect of the number of teacher models. For example, would it be a problem to distill 20 models? Especially when some styles are overlapped or similar.

**Questions:**

- What is the base models in Figure 5?
- In the experiment, the styles from different models are limited and overlapped, like realistic and anime, will it be a problem?
- Instead of merging base model directly, a more practical task is to merging multiple LoRAs, like style or character LoRAs, is this method applicable?

---

> ### Author Response · Authors · 2024-11-22
>
> **Thanks for valuable feedback**
>
> We appreciate the time you took to share your valuable feedback with us.
> For the visual demonstration, we provide image results in the **Supplementary Material PDF (cause we can not directly submit figs on openreview).**
>
> **Q.1 Concern about application.**
>
> There is indeed a demand for model merging techniques within the community; in fact, this technology is needed in our industrial products.
> Even though there are LoRAs available to achieve lightweight models, achieving satisfactory strong performance still requires full parameter fine-tuning and most top powerful models are still in the form of base models in the communities.
> In situations, we exactly should use the satisfying models and only have access to the base model (maybe the LoRAs are merged into the UNet), our approach is proposed to address this issue.
> Besides, the frequently used weighted merging is also trying to achieve the target but it is not flexible enough, so the application of our approach is desirable among the community.
>
> **Q.2 Necessity of model merging.**
>
> As mentioned above, the situation our method aims to address is the desire to flexibly and efficiently utilize existing expert models in the community, as these expert models are highly valuable due to being fine-tuned on high-quality datasets and can accurately satisfy our application requirements.
> The SDXL can cover many styles but it is too generalized and coarse, causing it not able to meet the needs of practical or even commercial applications.
>
> Compared to specific expert models, prompt tuning alone is very inefficient and falls far short of meeting application requirements and we conduct experiments to illustrate this.
> As shown in **Figure 3 in the PDF**, we try to control the style by adding keywords on SDXL, but the results are far from reaching the performance of our approach and other advantages of our DMM like style mixing are completely impossible.
>
> To put it bluntly, these fine-tuned models are indeed prettier than SDXL in practice, and in many cases, we exactly need to leverage these ready-made models for reasons like their high quality, unique strengths, etc.
> After all, they are carefully fine-tuned by users and possess domain-specific knowledge, making them more practical and effective compared to a general base model.
>
> **Q.3 Clarification for Table 3**
>
> Table 3 provides a perspective on the capabilities of our model, aiming to demonstrate that our approach can achieve results comparable to the original common practices by performing mix augmentation within similar domains.
> More investment can bring more advantages.
> First, our DMM can faithfully reproduce the effects of every teacher model, which further expands its versatility.
> Second, our approach can gather knowledge from many distinct models and apply flexible adjustment at inference time depending on the current need, while weighted merging cannot be adjusted after fixing the weights.
>
> **Q.4 Analysis of the number of teacher models**
>
> Sincerely thanks for your insightful suggestion for our approach principles as this determines the scalability of our DMM.
> We conducted experiments with three different settings for the number of teacher models, as shown in the table below.
> It should be acknowledged that the FID for each target slightly increases with the number of teacher models.
> Fortunately, this performance degradation is still quite manageable within a larger scope, at around 1\% every addition of about 10 models.
> This guarantees that our method can be effectively scaled to tens of teacher models.
>
> |#Models|1|2|3|4|5|6|7|8|9|10|11|12|13|14|15|16|
> |-|-|-|-|-|-|-|-|-|-|-|-|-|-|-|-|-|
> |4|9.5|9.8|9.7|9.5|-|-|-|-|-|-|-|-|-|-|-|-|
> |8|9.6|9.8| 9.6| 9.9| 9.8| 9.0| 9.8| 10.0|-|-|-|-|-|-|-|-|
> |16|9.6| 9.9| 9.7| 9.9| 9.9| 9.1| 9.8| 10.1| 9.9| 9.8| 9.8| 9.9| 9.5| 10.0| 9.9| 9.8|
>
> **Q.5 What is the base models in Figure 5?**
>
> The information on all the teacher models is provided in the Appendices.
> As stated in the manuscript, we merge the first four styles, namely the base models are the first four models in Table 4 in the main paper.
>
> **Q.6 Concern about style overlap**
>
> This will not cause problems.
> Similar styles will only simplify the training task, and even models with similar styles will have subtle differences.
> Our approach is capable of learning the specific knowledge and faithfully reproducing each teacher model.
>
> Additionally, we conduct experiments on more special-style models as demonstrated **in Section Q.18 in the PDF**, and our approach still works well.
>
> **Q.7 Applicability of merging LoRAs**
>
> Yes, it is feasible to apply our method for merging LoRAs.
> In fact, in the above experiment **in Section Q.18 in the PDF** of unique style, the teacher models are in the form of LoRAs, and we have successfully merged them. We can also implement the trainable parameters as LoRA.

---

> > ### Comment · Reviewer_SqM4 · 2024-11-30
> > **Response to Authors**
> >
> > Thanks for response, it solves part of my concerns. But I still have the following questions
> >
> > - Still about the application scenarios, for SD1.5/SDXL, indeed, there are many base models (lora-merged or finetuned) in the community. However, as the model size continues to increase, it becomes increasingly difficult to train or fuse the base model for SD3.5/FLUX.1. Instead, the community prefers to learn the style in the form of LoRA. In this case, the advantage of DMM is not obvious compared to training-free weighted merging, especially the consumption of VRAM.
> > - There are a limited number of specialized models, which usually fall into a few common categories such as realistic models or anime models. Since the author also mentioned that DMM can be used to fuse LoRA, in addition to style fusion, I am curious about the effect of the fusion of character LoRAs and whether the models will interfere with each other, especially when the quality of different LoRAs is different.

---

> > > ### Author Response · Authors · 2024-12-03
> > >
> > > Thank you for your response. We hope to relieve your remaining concerns:
> > > - Despite SD3.5/FLUX.1 and LoRA, they still face the problem of efficiently assembling multiple powerful expert models and we have illustrated that our approach can be applied to LoRAs. We have highlighted the limitations and inconveniences of weighted merging. For example, (1) it is an offline method and cannot be adjusted online flexibly, (2) multi-style conflicting, which cannot satisfy many practical applications. The investment in training can indeed yield benefits that far surpass those of this naive method and sometimes we exactly need these.
> > > We believe that our method explores new natures about model merging in the T2I community and provides a valuable baseline option for users who wish to invest resources in developing applications.
> > >
> > > - Please refer to **in Section Q.18 and Q.24 in the PDF**. We have thoroughly validated that our method is effective not only across the few common styles but also in a broader range of domains, including content, quite special and different styles, detail preferences, and even conditional control. Our method aims to learn different knowledge independently. The many results have shown that different components will not interfere with each other and can form a smooth transition. The character LoRAs have no difference from the presented models in essence. We will conduct such experiments and provide it in the revised paper, as now the PDF can not be updated

---

### Official Review · Reviewer_Za1P · 2024-11-07

**Soundness:** 2
**Presentation:** 2
**Contribution:** 2
**Rating:** 5
**Confidence:** 3

**Summary:**

This paper proposes a distillation-based model merging technique to learn a single model that can mimic the styles of N teacher models. To that end, learnable style prompts are used that trigger the generation of the style of a specific teacher model. The student is trained to reconstruct the teacher generations using a feature similarity loss (feature imitation), noise reconstruction (score distillation), and a multi-class adversarial loss. Experiments include learning a single model from 8 teacher styles and a continual learning setup.

**Strengths:**

- Overall, this paper presents a compelling narrative. It begins by addressing a well-motivated problem: the need to merge multiple specialized models into a single model to reduce computational demands in text-to-image generation.
- The literature review and introduction is thorough, with a well-structured connection between various fields.
- The paper introduces a combination of techniques to improve model merging performance, and combining these approaches yield the best results.
- The proposed evaluation metric is appropriate, as it involves evaluating FIDs across separate tasks and styles.
- The paper includes numerous visual results that support the capabilities of the trained model.
- Additionally, the descriptions of the proposed methods are clear, and the figures are well-presented.

**Weaknesses:**

My biggest concern is w.r.t. made claims regarding problems of naive merging or other existing methods, lack of strong visual and quant. evidence for improving over the baseline, and the setup of the teacher models.

- The selection of 8 styles, with 4 styles categorized as "realistic" and 2 as "anime" is a weak setup to effectively demonstrate the problem as well as the advantages of the proposed model.
- The demonstration of the issue being addressed and the impact of the proposed solution is missing / low.
  - Claims regarding issues with simple merging lack supporting evidence. Specific conflicts, ambiguities, or patterns of confusion are not demonstrated.
  - In Table 1, baseline results without any proposed additions (e.g., the initialized student model) are missing and would provide more context.
  - Table 2 would benefit from including upper-bound teacher performance for comparison.
  - Visual results for baseline and ablation configurations are missing, which are critical to substantiating claims of improvements and understanding the impact of reducing FID from 80 to 77.
- The scope is limited to style transfer; extending the experiments to different content domains (e.g., mixing human faces with objects or indoor and outdoor scenes) would strengthen the argument.
- The discussion on continual learning appears unrelated to the core problem of model merging and feels out of place.
- Reporting individual FIDs instead of averages in the continual learning setup would provide more insights.
- In Table 3, the proposed model achieves basically the same performance, but text and results lacks evidence for benefits beyond claimed flexibility. The method for manually determining weights in weighted merging is unclear, as is the sensitivity of this process compared to naive interpolation.
- The approach still requires access to the fine-tuned models (at least during training), which limits its practicality.
- The method for determining weights for the total loss and their sensitivity is not adequately explained.
- The term DDPM should be fixed as *Denoising Diffusion Probabilistic Model* rather than "Probability".
- Generative Adversarial Networks is the correct term (attributed to Goodfellow et al.), not "Generative Adversary Networks" (incorrectly attributed to Song & Ermon).

**Questions:**

- Clarification is needed on the exact nature of the style prompts: Are these special text tokens appended to the prompts, or is something else trainable in the student model? A similar results might be achievable by tuning prompts and using style-related keywords, as shown by prior work on prompt tuning.

---

> ### Author Response · Authors · 2024-11-22
>
> **Thanks for the valuable feedback**
>
> We appreciate the time you took to share your valuable feedback with us.
> For the visual demonstration, we provide image results in the **Supplementary Material PDF** (cause we can not directly submit figs on openreview).
>
> **Q.1 The selection of styles is a weak setup to demonstrate the problem.**
>
> To further demonstrate the generalizability of our approach, we conduct experiments on a wider variety of distinct models, including Chinese ink painting, pixel art, business illustration, etc. We show the results in Section Q.18 in the PDF. Our DMM still works well on these challenging models.
>
> **Q.2 Demonstration of issues being addressed**
>
> To concretely demonstrate that directly merging models of various styles will cause conflict and style confusion, we conduct such experiments. With the same setting of model candidates, namely the eight models with different styles, we apply weighted merging directly on their parameters and use the merged model to synthesize images.
>
> **As shown in Figure 1 (in the PDF), the output of this model exhibits an ambiguous hybrid style, rather than being purely realistic or entirely anime style.**
> Even with additional corresponding keywords (e.g. *'realistic', 'animation'*), it still struggles to accurately distinguish between different styles and cannot produce satisfactory results, compared to our results in Figure 6 in the main paper.
> This result is not what we expected and we cannot effectively control the generation.
>
> This phenomenon can also be used to explain why prompt engineering cannot achieve the same effects as our merging approach, since handcraft prompt tuning operates at a relatively vague level, while our method can provide precise suggestions and access expert model knowledge.
>
> **Q.3 Lack of baseline results in Table 1**
>
> We use the **initialized student model**, namely SDv1.5 as the baseline results. Specifically, for the FID calculation of each style, we use additional style-related keywords in the text prompt to modulate different styles.
> It is obvious that the baseline results are very poor, and the visualizations below will demonstrate this point again.
>
> |Methods|FIDt$\downarrow$|
> |----------|---------------------|
> |Baseline|195.22|
> |+ Score Distillation| 80.69 |
> |+ Feature Imitation| 79.27 |
> |+ Multi-Class Adversarial | 78.38 |
> |+ Synthesized Finetune | **77.51** |
> |Teacher Reference| 74.91 |
>
> **Q.4 Missing upper-bound teacher performance in Table 2**
>
> We will provide the upper-bound results in Table 2 in the final revision.
>
> **Q.5 The scope is limited**
>
> We further extend the application scope of our DMM approach:
> - As mentioned above, we conduct experiments on merging more challenging models, which have both different styles and different contents (Chinese ink painting, outdoor scenes, etc.). The results are provided **in Section Q.18 in the PDF**.
> - We conduct experiments on merging different control conditions by applying our approach to several ControlNets[1]. It works well and can combined with DMM-UNet feasibly, achieving versatile styles and versatile controls orthogonally with only a single model. The results are provided **in Section Q.24 in the PDF**.
>
> **Q.6 Missing visual results for baseline and ablation configurations.**
>
> We provided visual results in **Figure 2 (in the PDF)** of different ablation settings to verify the improvement in visual quality brought about by the reduction in FID.
> It can be seen that, as the FID decreases, the images exhibit more refined optimizations, such as improved textures and lighting, indicating a closer alignment with the target style.
>
> **Q.7 Continual learning appears unrelated to the core problem**
>
> That is not the case. The sustainability of our method is crucial for the application of model merging and should be paid attention to.
> If our approach is a disposable solution, its application flexibility would be greatly limited.
> Users often encounter situations where they want to incrementally merge a recently developed model into the current checkpoint. If they have to train from scratch, it will bring inconveniences and resource waste, while our continual learning strategy alleviates the problem.
>
> **Reference**
>
> [1] Zhang, Lvmin, Anyi Rao, and Maneesh Agrawala. "Adding conditional control to text-to-image diffusion models." Proceedings of the IEEE/CVF International Conference on Computer Vision. 2023.

---

> > ### Author Response · Authors · 2024-11-22
> >
> > **Q.8 More insights from Table 2 result in continual learning.**
> >
> > Thanks for this suggestion and we deeply analyze the result in Table 2.
> > First, considering experiments #1 and #2, we can observe that for the first four FIDs, the results of #2 are slightly worse than those of #1.
> > This potentially hints at a scaling relationship between performance and the number of models, and it is understandable, as the increase in the number of models would challenge the overall capacity and performance of the models.
> > Fortunately, this performance degradation is still quite manageable within a larger scope, at around 1%.
> > Second, considering #3 and #4, freely fine-tuning the last four models in #3 shows a slight advantage compared to #4, but introduces unacceptable bursts in the first four FIDs.
> > It is desirable to employ our proposed regularization method to achieve balanced results.
> > Besides, these insights will also inspire us to analyze the scaling laws regarding the number of models and the performance, which we conduct in Q.14 in the PDF.
> >
> > **Q.9 Clarification of results compared to weighted merging in Table 3**
> >
> > In the first place, Table 3 provides one perspective on the capabilities of our model, aiming to demonstrate that our approach can cover the effects of original common practices by performing mix augmentation within similar domains.
> > In addition, our method **offers more advantages in terms of flexibility** compared to weighted merging, such as faithful reproduction of teacher models, easier weight tuning at inference time, and compatibility with many other frameworks.
> >
> > As for the determination of the weights, the optimal weights should be searched manually for both methods.
> > However, the original method is disposable since the parameters cannot be adjusted once the weight-merged model file is dumped.
> > In contrast, our method allows for free weight control during inference time through input arguments based on current requirements, making it much more flexible.
> >
> > **Q.10 Requires access to the fine-tuned models**
> >
> > The goal of the model merging task is to integrate multiple fine-tuned models, which naturally require access to them.
> > This is the same in many other task fields, such as the merging of LLMs, detection models, etc.
> > We are truly addressing practical issues in the T2I community: **users exactly want to utilize these ready-made expert models more efficiently and flexibly.**
> >
> > **Q.11 Typos for the term "DDPM" and "GAN"**
> >
> > We apologize sincerely for the typos, and thoroughly checked and rectified them. Thanks for pointing out these typos.
> >
> > **Q.12 Nature of the style prompts**
> >
> > We have provided the implemented details of style prompts in the Appendices. Sorry for not making it clear, we will place this important information in the main text rather than in an appendix.
> > Specifically, we first align the embeddings with an MLP and then add it to the timestep embedding, which is lightweight enough for plug-and-play purposes.
> > The shape of style prompts is $(N, 1280)$, where $N$ is the number of models.
> > The MLP contains two $(1280, 1280)$ linear projection.
> > That is all the additional trainable parameters.
> >
> > **Q.13 Similar results can be achieved through prompt tuning?**
> >
> > We illustrate this **in Section Q.12 in the PDF**, where we leverage the powerful SDXL to generate distinct styles by adjusting the prompts.
> > Unfortunately, the prompt tuning method is quite inefficient and completely cannot yield the expected results.
> >
> > Notably, the core issue we want to address is to efficiently utilize expert models, which is significantly needed in the community.
> > After all, they are carefully fine-tuned by users and possess domain-specific knowledge, making them more practical and effective compared to a general base model.

---

> ### Author Response · Authors · 2024-11-27
>
> Dear review, as the revised PDF upload ends on November 27, we want to ensure that all your questions have been thoroughly addressed. Your feedback is instrumental to us, and we would be grateful if you could spare a moment to provide a final rating and share your thoughts. Your input will greatly inform our future improvements.

---

### Author Response · Authors · 2024-11-22

We thank all reviewers' efforts in reviewing our paper and giving insightful comments and valuable suggestions.
We provide **a PDF file of the content of our rebuttals in Supplementary Material, to include many visual demonstrations.**

If the reviewers have any constructive feedback or suggestions for further improvement, please don't hesitate to reach out to me directly.

---

### Author Response · Authors · 2024-11-25

Dear reviewers, as the discussion period ends on November 26, we are eager to ensure that all the questions have been thoroughly resolved. We hope that our responses have adequately addressed your concerns. Your feedback is invaluable to us, and we would greatly appreciate it if you could take a moment to provide a final rating and feedback.

---

### Meta-Review · Area_Chair_d25s · 2024-12-17

**Metareview:**

This paper presents a score distillation approach to merge multiple models into a single T2I model capable of generating images with arbitrary styles. The paper was reviewed by four knowledgeable referees. The reviewers acknowledged that the problem was motivated (Za1P, SqM4), and the proposed pipeline reasonable (SqM4).

The reviewers also raised concerns, summarized as:
1. Unconvincing experimental evidence to support the claims (Za1P, JLMo), with a limited scope of the experimental results (Za1P, V6za, SqM4), and distillation from multiple models missing (JLMo)
2. The unclear advantage and applicability of the proposed method (SqM4)
3. The performance gains appearing minor w.r.t. weighted merging (SqM4)

There were also questions about demonstrating the ability to deal with catastrophic forgetting (V6za).

The rebuttal partially addressed the reviewers' concerns by presenting experiments trying to demonstrate the problem being addressed, adding some baseline results, experiments with increased numbers of target models, and incremental learning experiments. The rebuttal also argued the relevance of the problem tackled in industrial settings. However, after the discussion period, the reviewers remained unconvinced. In particular, the reviewers kept raising concerns about the unconvincing demonstration of the problem this paper wants to solve and the effectiveness of the proposed approach (benefits / improvements of the proposed approach w.r.t. existing baselines and missed consistent comparisons with the baselines). Therefore, after discussion, all reviewers lean towards rejection. The MR agrees with the reviewers' assessment and therefore recommends to reject. The MR encourages the authors to consider the reviewers' feedback to improve future iterations of their work, including potential user studies to highlight the benefits of their proposed approach.

**Additional Comments On Reviewer Discussion:**

See above.

---

### Decision · Program_Chairs · 2025-01-22

Reject